# Exploiting oxidative phosphorylation to promote the stem and immunoevasive properties of pancreatic cancer stem cells

Sandra Valle[1,2], Sonia Alcalá [1,2], Laura Martin-Hijano[1,2], Pablo Cabezas-Sáinz [3], Diego Navarro[1,2], Edurne Ramos Muñoz[4], Lourdes Yuste[1,2], Kanishka Tiwary [5], Karolin Walter[5], Laura Ruiz-Cañas[1,2], Marta Alonso-Nocelo [1,2], Juan A. Rubiolo [3], Emilio González-Arnay[6], Christopher Heeschen[7,11], Laura Garcia-Bermejo[4], Patrick C. Hermann[5], Laura Sánchez [3], Patricia Sancho[8], Miguel Ángel Fernández-Moreno [1,9,10 ✉] & Bruno Sainz Jr [1,2 ✉]

Pancreatic ductal adenocarcinoma (PDAC), the fourth leading cause of cancer death, has a 5-year survival rate of approximately 7–9%. The ineffectiveness of anti-PDAC therapies is believed to be due to the existence of a subpopulation of tumor cells known as cancer stem cells (CSCs), which are functionally plastic, and have exclusive tumorigenic, chemoresistant and metastatic capacities. Herein, we describe a 2D in vitro system for long-term enrichment of pancreatic CSCs that is amenable to biological and CSC-specific studies. By changing the carbon source from glucose to galactose in vitro, we force PDAC cells to utilize OXPHOS, resulting in enrichment of CSCs defined by increased CSC biomarker and pluripotency gene expression, greater tumorigenic potential, induced but reversible quiescence, increased OXPHOS activity, enhanced invasiveness, and upregulated immune evasion properties. This CSC enrichment method can facilitate the discovery of new CSC-specific hallmarks for future development into targets for PDAC-based therapies.

[1] Department of Biochemistry, Universidad Autónoma de Madrid (UAM) and Instituto de Investigaciones Biomédicas "Alberto Sols" (IIBM), CSIC-UAM, Madrid, Spain. [2] Cancer Stem Cells and Fibroinflammatory Microenvironment Group, Chronic Diseases and Cancer Area 3 - Instituto Ramón y Cajal de Investigación Sanitaria (IRYCIS), Madrid, Spain. [3] Department of Zoology, Genetics and Physical Anthropology, Veterinary Faculty, Universidad de Santiago de Compostela, Lugo, Spain. [4] Biomarkers and Therapeutic Targets Group - IRYCIS, Madrid, Spain. [5] Department of Internal Medicine I, Ulm University, Ulm, Germany. [6] Department of Neurology, Universidad Autónoma de Madrid (UAM), Madrid, Spain. [7] Stem Cells & Cancer Group, Molecular Pathology Programme, Spanish National Cancer Research Centre (CNIO), Madrid, Spain. [8] Translational Research Unit, IIS Aragón, Zaragoza, Spain. [9] Centro de Investigación Biomédica en Red en Enfermedades Raras (CIBERER), Madrid, Spain. [10] Instituto de Investigación Sanitaria Hospital 12 de Octubre (imas12), Madrid, Spain. [11] Present address: Center for Single-Cell Omics and Key Laboratory of Oncogenes and Related Genes, Shanghai Jiao Tong University School of Medicine, Shanghai, China. ✉email: miguel.fernandez@uam.es; bsainz@iib.uam.es

The past decade has seen great progress in the diagnosis and treatment of different cancers; however, the same cannot be said for pancreatic ductal adenocarcinoma (PDAC), which is currently the fourth most frequent cause of cancer-related death and projected to become the second most lethal tumor by the year 2030[1]. At the time of diagnosis, <20% of patients present with localized (and thus potentially resectable and curative) disease, 15–20% of patients have locally advanced (unresectable) tumors, and the remaining patients present with metastatic disease. To compound the situation further, even the strongest chemotherapeutic regimens only extend overall survival to approximately 11 months and very rarely result in long-term progression-free survival (>5 years)[2]. Thus radically new approaches are needed to identify novel and more effective therapies for PDAC.

To achieve the latter, it is important to appreciate that pancreatic tumors are comprised of a heterogeneous population of cancer cells with diverse replicative, tumorigenic, metastatic, and chemoresistant capacities. Specifically, highly plastic sub-populations of stem-like cells within the tumor, known as cancer stem cells (CSCs), have been described for PDAC and are now accepted to be the drivers of tumorigenesis, chemoresistance, and metastasis[3]. Using CSC surface markers or three-dimensional (3D) spheroid cultures to isolate pancreatic CSCs (PaCSCs), we have advanced our ability to study and molecularly characterize these cells. For example, we now appreciate that PaCSCs are highly plastic and therefore have the capacity to alter their metabolism, induce quiescence, resist chemotherapeutic insults, and even evade the immune system by upregulating immune evasion receptors or checkpoint inhibitors[4]. While much progress has been made, we are still far from completely understanding the complex biological makeup of PaCSCs, largely due to the inability to establish long-term two-dimensional (2D) cultures enriched in CSCs that are amenable to characterization and screening studies.

Approaches that use specific signaling mechanisms to enrich and maintain CSC subpopulations with phenotypic plasticity in culture[5] could prove useful for (1) dissecting the underlying drivers of their plasticity and for (2) identifying new anti-CSC targets for drug development. Toward this end, and based on our previous work demonstrating the metabolic differences that exist between PaCSCs and non-PaCSCs, we describe herein a long-term and sustained cell culture system enriched in PaCSCs based on forced oxidative phosphorylation (OXPHOS), defined by electron transport chain + ATP synthase. In this culture system, cells increase the expression of PaCSC biomarkers and present better metabolic adaptation, plastic features such as a reversible quiescence-like state, and CSC-associated phenotypes, including chemoresistance, invasiveness, and immune evasive properties. This culture method not only sheds light on the link between previously unrecognized CSC features and mitochondrial-dependent metabolism but also represents a platform that could facilitate the discovery of CSC-specific properties that could lead to the development of new therapies against PaCSCs, which could ultimately improve the life expectancy of PDAC patients.

## Results

**PaCSCs have increased mitochondrial function**. While 3D anchorage-independent spheres allow for CSC enrichment, they are not highly adaptable to various screening methodologies due to their non-adherent 3D nature and need for serial passaging once they reach critical mass. Attempts to establish 2D cultures enriched in CSCs using fluorescence-activated cell sorter (FACS) approaches have also proven ineffective as marker-enriched CSCs quickly re-establish the heterogeneity of the culture present prior to sorting. Thus the enrichment in PaCSCs is not a stable phenotype and marker-enriched [CD133 and autofluorescence (Fluo)[6,7]] CSCs quickly return to their pre-sort distribution within 2–3 days post sorting (Fig. 1a). For these reasons, we decided to further dissect CSCs to discover other properties that could facilitate a long-term CSC-enriched 2D culture. Using the classical PaCSC biomarker CD133, we divided PaCSCs (CD133+) from non-PaCSCs (CD133−) and carried out a transmission electron microscopy (TEM)-based analysis and observed morphological and structural difference in their mitochondria. CD133+ cells (PaCSCs) contained more differentiated mitochondria compared to CD133− cells (non-PaCSCs), which exhibited swollen mitochondria with less pronounced cristae and which appeared to be eliminated by autophagy (Fig. 1b). To validate these observations, PaCSCs were sorted and marked with probes for mitochondrial mass/membrane potential (MTDR and MT-CMXRos). We observed that CD133+Fluo+ cultures showed a significant increase in MTDR of ~3–4-fold and a significant increase in MT-CMXRos of ~2–3-fold compared to CD133−Fluo− cells (Fig. 1c–e). Quantification of the mitochondrial 12S ribosomal RNA gene (MT-RNR1) also revealed a significant increase in the number of mitochondrial DNA (mtDNA) copy numbers in the CD133+Fluo+ population (Fig. 1f). These data confirmed previous studies showing that PaCSCs have a more active mitochondrial state[8,9], which we reasoned could be exploited to establish a long-term CSC-enriched 2D culture. Toward this end, we used 5 mM galactose as an alternate carbon source in our culture medium. Compared to 5 mM glucose, ATP yield from 5 mM galactose is slower due to the fact that the conversion of galactose to glucose 6-phosphate requires several more limiting enzymatic steps (Leloir pathway)[10], and as shown by Warburg in 1925, cancer cells use 18-fold less glycolysis when galactose, in lieu of glucose, is provided as a sugar source[11]. Thus galactose, together with exogenous mitochondrial OXPHOS substrates (i.e., pyruvate and glutamine) leads to a compensatory upregulation of OXPHOS[12]. As a consequence, cells that are dependent on glycolysis but not on OXPHOS function (non-PaCSCs) cannot survive in galactose, while cells that have an efficient OXPHOS system (PaCSCs) will survive and become enriched (Fig. 1g).

**Galactose-cultured cells are enriched in PaCSCs**. First, using the IncuCyte Zoom System we performed a real-time analysis of the CSC marker autofluorescence and cell confluency for PANC185 cells during 40 h of culture in glycolytic-restrictive conditions (e.g., galactose). Interestingly, relative confluency was static and did not change in galactose-cultured cells (Gal-CC) compared to glucose-cultured cells (Gluc-CC) (Fig. 2a), but the ratio of the CSC autofluorescent biomarker/relative confluency increased ~2-fold in Gal-CC (Fig. 2b), indicating an enrichment in the CSC population. Enrichment in autofluorescence and reduction in cell confluency (i.e., proliferation) was more pronounced at longer times (14 days) (Supplementary Fig. 1a, b), and coincided with increased apoptosis in Gal-CC versus Gluc-CC across 4 PDAC cell lines (Supplementary Fig. 1c), indicating that the observed enrichment in PaCSCs was likely a consequence of cell death and inhibition of cell proliferation. To validate the observed enrichment in PaCSCs, we performed an extensive flow cytometry-based (Supplementary Fig. 1d) analysis using other CSC biomarkers[6,13]. Importantly, we observed that autofluorescence, CD133, and CD24 significantly increased by more than twofold and CXCR4 and TEM8 significantly increased by threefold in 14-day Gal-CC compared to Gluc-CC (Fig. 2c, d). Moreover, CD90 and SSEA-4, PaCSC biomarkers present only on PANC286 and PANC185scd, respectively, significantly increased by more

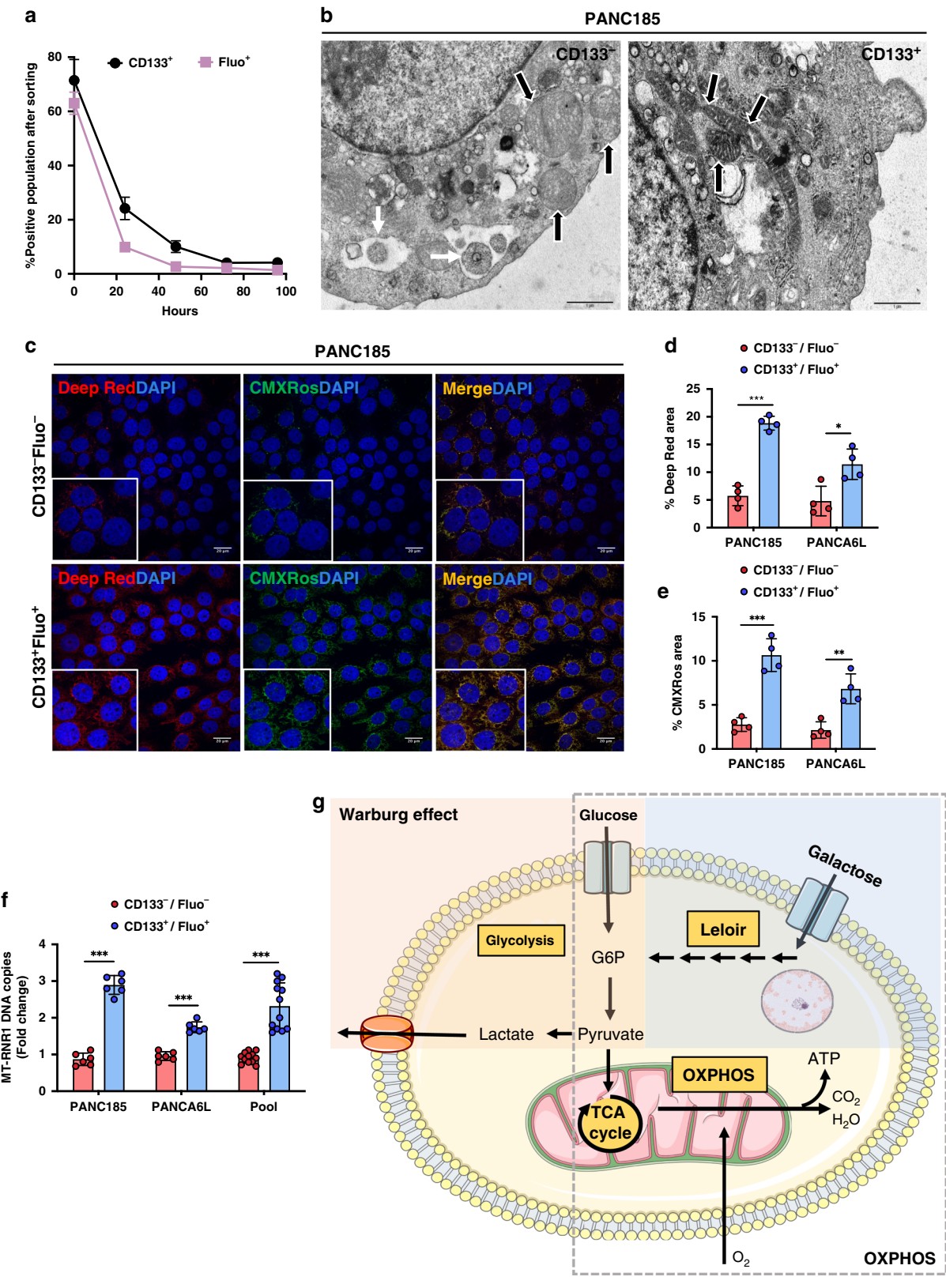

than fourfold (CD90) or twofold (SSEA-4) in 14-day Gal-CC (Supplementary Fig. 1e, f). Maximum enrichment in PaCSC marker expression in Gal-CC was observed at 14 days and thus this time point was used for all experiments presented here onwards. Importantly, the levels of CSC biomarker enrichment in Gal-CC versus Gluc-CC were greater than those obtained for

sphere versus adherent cultures (Supplementary Fig. 1g). Lastly, long-term PaCSC-enriched Gal-CC could be established directly from a freshly resected patient-derived xenograft (PDX) tumor (Supplementary Fig. 1h, i), highlighting the translatability of these findings and the putative applicability of this system to surgically resected patient tumors.

**Fig. 1 PaCSCs have increased mitochondrial function. a** Primary cultures of PANC185 cells were sorted using the CSC markers CD133 or autofluorescence (Fluo), and the percentage of CD133+ (black) or Fluo+ (purple) cells was re-determined by flow cytometry at the indicated hours post sorting ($n = 3$ biological replicates). **b** PANC185 cells were stained with an anti-CD133-APC antibody and sorted to divide the cell population into CD133− cells (non-CSCs) and CD133+ cells (CSCs). Transmission electron micrographs were acquired and shown are representative images. Mitochondria (black arrow) are better defined in the CD133+ population compared to CD133−, which present autophagosomes (white arrow) in their cytoplasm. **c** PDAC cultures were sorted using two different CSC biomarkers, CD133 and autofluorescence (Fluo) to select two cell populations: CD133−Fluo− cells (non-CSCs) and CD133+ Fluo+ cells (CSCs). Five days post re-seeding, both populations were marked with the mitochondrial markers MitoTracker-DeepRed FM (red) and Mitotracker-CMXRos (green), and DAPI (nuclear marker, blue). Shown are representative fluorescent images obtained by confocal microscopy analysis for PANC185. **d, e** Quantification of the percentage of mean fluorescent area ± sd for the **d** MitoTracker-DeepRed FM mitochondrial biomarker or the **e** Mitotracker-CMXRos mitochondrial membrane potential marker in PANC185 and PANCA6L, using ImageJ ($n = 4$ independent fields; Holm–Sidak $t$ test statistical analysis). DeepRed ***$p = 0.00004$; *$p = 0.013$ and CMXRos ***$p = 0.00045$; **$p = 0.0028$. **f** Quantification of mean relative mRNA expression levels ± sd of the mitochondrial 12S ribosomal RNA gene (MT-RNR1) by qPCR in CD133+Fluo+ cells and CD133−Fluo− cells in PANC185 and PANCA6L ($n = 6$ biological replicates or $n = 12$ biological replicates for Pooled data; Holm–Sidak $t$ test statistical analysis). Data were normalized to β-actin expression and presented as fold change for each tumor and pooled data. CD133−Fluo− was set as 1.0. ***$p < 0.000001$; ***$p = 0.000001$; ***$p < 0.000001$. **g** Schematic representation of the Warburg and OXPHOS metabolism present in glucose and galactose culture conditions.

While the above results supported that cell death and inhibition of cell proliferation in Gal-CC was responsible for PaCSC enrichment, these data could not negate the additional possibility that CSC biomarker-negative cells were converting into CSC biomarker-positive cells and also contributing to the observed enrichment in PaCSCs. To address the latter, a double FACS was performed to obtain highly pure CD133- or autofluorescent-negative cell populations, which were then cultured in glucose- or galactose-containing media to assess the plasticity of CSC biomarker-negative cells. Independent of the carbon source, CSC biomarker-negative cells gave rise to CSC biomarker-positive cells, confirming previous observations regarding plasticity of non-CSCs (Supplementary Fig. 2a, b). However, at times >4 days post-sorting, a sharp decrease in the percentage of CSC biomarker-positive cells was observed in Gluc-CC, while in Gal-CC the percentages either remained constant or increased, and at day 11 were ~2-fold higher than Gluc-CC (Supplementary Fig. 2a, b), indicating that galactose culture conditions potentiate and maintain plasticity. Importantly, this fold difference coincided with an ~2-fold decrease in cell viability in Gal-CC, confirming that PaCSC enrichment also coincides with non-PaCSC cell death (Supplementary Fig. 2c). Importantly, when CD133+ sorted cells were directly cultured in galactose, cell viability was not negatively affected (Supplementary Fig. 2d). Taken together, we concluded that in galactose-containing media (1) PaCSCs survive, (2) a large percentage of non-PaCSCs die, and (3) a smaller percentage of non-PaCSCs convert into PaCSCs. The sum of these biological consequences results in a PaCSC-enriched culture.

**Gal-CC contain functional PaCSCs.** We next characterized Gal-CC at the molecular and functional level to confirm an enrichment in PaCSCs. First, we measured self-renewal capacity and observed significantly increased sphere-forming capacity for Gal-CC compared to Gluc-CC (Fig. 2e), although spheres from Gluc-CC were consistently larger (Fig. 2f). This increase over two generations mirrors what is seen when PaCSCs sorted for CD133 are placed in sphere-formation conditions (Supplementary Fig. 2e). Of note, we confirmed that the observed increase in self-renewal capacity in Gal-CC was due to an enrichment in the percentage of CSCs by showing that the percentage of autofluorescent-positive (Fig. 2g) and CD133+CXCR4+ double-positive (Fig. 2h) cells were significantly higher in Gal-CC (more than threefold) compared to Gluc-CC in first- and second-generation spheres. At the transcriptional level, and similar to what is observed when PaCSCs are enriched using sphere formation (Supplementary Fig. 2f), genes related to stemness (e.g., SOX9) and pluripotency (e.g., NANOG) were significantly

increased 2–3-fold in Gal-CC compared to Gluc-CC (Fig. 2i). The latter was validated using a NANOG-YFP reporter system, in which Yellow Nano-lantern (YNL) expression (indicative of hNANOG promoter activity) was significantly higher in Gal-CC compared to Gluc-CC (Fig. 2j). Recent evidence suggests that different types of non-coding RNAs, such as long non-coding RNAs (lncRNAs), also play a role in regulating CSC growth and replication during cancer growth and metastasis[14,15]. Analysis of the expression levels of a subset of lncRNAs revealed a significant 2–3-fold increase in their expression in Gal-CC compared to Gluc-CC (Supplementary Fig. 1g), similar to our previously published data using adherent and sphere cultures[16].

As tumorigenesis is a hallmark of CSCs, an in vivo extreme limiting dilution assay (ELDA) was performed using mCherry-H2B-labeled Gal-CC and Gluc-CC. While the size, weight, and percentage of mCherry-H2B+ Gluc-CC-derived tumors were slightly greater or equal to those of the Gal-CC-derived tumors (Supplementary Fig. 2h–l), tumor incidence for galactose-derived cells was significantly higher at all dilutions tested (Fig. 2k). Moreover, the percentage of CD133+ cells inside the mCherry-H2B+ cell population was significantly higher in the vast majority of tumors derived from Gal-CC versus Gluc-CC (Fig. 2l), indicating an expansion of the CSC compartment, which was confirmed by calculating the CSC frequency (Fig. 2m): Gal-CC (1 in 10,075) versus Gluc-CC (1 in 51,521), and in line with the frequencies achieved when using the CSC markers CD133 or autofluorescence to separate PaCSCs from non-PaCSCs (Supplementary Fig. 2m). Taken together, the sum of these data indicates that there exists an enrichment in CSCs when cells are cultured in conditions that demand OXPHOS.

**PDAC cells increase OXPHOS in the presence of galactose.** We previously published that PaCSCs use OXPHOS to meet their energy requirements[8]. Likewise, Viale et al. showed that CSC-like cells that resist oncogene ablation in PDAC murine models rely on OXPHOS for their survival[17]. Since Gal-CC should be more OXPHOS permissive, mitochondrial metabolism, markers of mitochondrial mass/membrane potential and reactive oxygen species (ROS) were analyzed (Fig. 3a). In line with our approach, all markers significantly increased in Gal-CC (Fig. 3b), suggesting increased OXPHOS and ROS production. Since some of the aforementioned probes depend on membrane potential, we additionally tested MitoTracker Green and Nonyl Acridine Orange (NAO) and observed a similar significant increase in mitochondrial mass (Supplementary Fig. 3a, b). To functionally validate these findings, we measured the oxygen consumption rates (OCRs) of Gluc-CC and Gal-CC (Fig. 3c). Baseline OCR was significantly higher in Gal-CC compared to Gluc-CC, and both

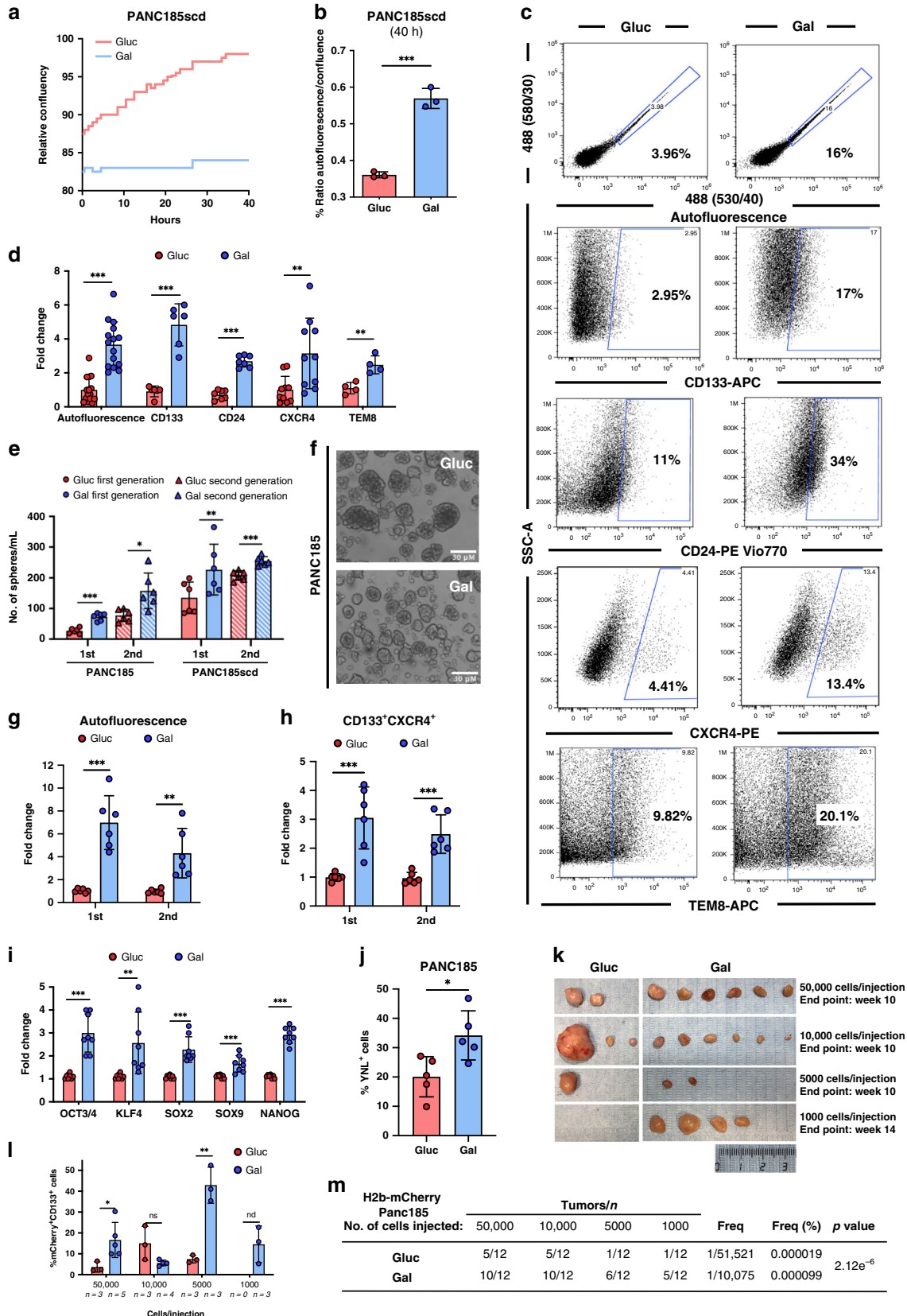

the maximal respiration and the spare respiratory capacity (SRC) significantly increased in Gal-CC compared to Gluc-CC (Fig. 3c, d), indicating increased OXPHOS. Proton leak was also significantly higher in Gluc-CC, indicating decreased mitochondrial efficiency (Fig. 3d). As expected, extracellular acidification rate (ECAR) revealed more acidification by glucose fermentation in Gluc-CC compared to Gal-CC (Supplementary Fig. 3c), highlighting that OXPHOS is preferentially used over glycolysis in Gal-CC. While direct confirmation of increased carbon flux into the mitochondria (i.e., tricarboxylic acid cycle) would need to be

**Fig. 2 Gal-CC are enriched in PaCSCs. a** Real-time quantification of cell confluence. **b** Quantification of mean ratio ± sd of the percentage of autofluorescence/confluence at 40 h, extrapolated from the data acquired from the IncuCyte Zoom System. ($n = 3$ biological replicates; Holm–Sidak $t$ test statistical analysis). ***$p = 0.00022$. **c** Representative flow cytometric plots of the percentage of CSCs biomarkers. **d** Quantification of mean fold change ± sd of CSCs biomarkers for 4 different tumors (biological replicates: $n = 14$ for autofluorescence; $n = 6$ for CD133; $n = 7$ for CD24, $n = 10$ for CXCR4; $n = 4$ for TEM8; Holm–Sidak $t$ test statistical analysis). Gluc was set as 1.0. ***$p < 0.000001$; ***$p = 0.00002$; ***$p < 0.000001$; **$p = 0.0067$; **$p = 0.0045$. **e** Mean number (no.) ± sd of first- and second-generation (gen) spheres/ml at 7 and 14 days ($n = 6$ biological replicates; Holm–Sidak $t$ test statistical analysis). ***$p = 0.000025$; *$p = 0.043$; **$p = 0.0094$; ***$p = 0.00029$. **f** Representative PANC185 sphere images at 14 days. Scale bar = 30 µM. **g, h** Flow cytometric quantification of mean fold change ± sd of **g** autofluorescent-positive or **h** CD133+CXCR4+ cells detected in spheres at 7 and 14 days. Gluc was set as 1.0. ($n = 6$ biological replicates; Holm–Sidak $t$ test statistical analysis) For **g**: ***$p = 0.0001$; **$p = 0.0036$. For **h**: ***$p = 0.0008$; ***$p = 0.0003$. **i** RT-qPCR analysis of relative mRNA expression levels of different pluripotency/stemness genes for 4 different tumors. Levels are normalized to β-actin ($n = 2$ biological replicates for each tumor; Holm–Sidak $t$ test statistical analysis). Data are presented as pooled mean fold change ± sd. Gluc was set as 1.0. ***$p = 0.000017$; **$p = 0.0085$; ***$p = 0.000034$; ***$p = 0.00085$; ***$p < 0.000001$. **j** Flow cytometric quantification of the mean percentage ± sd of Nanog reporter YFP+ PANC185 cells at 14 days. ($n = 5$ biological replicates; Mann–Whitney $t$ test statistical analysis). *$p = 0.032$. **k** Images of tumors obtained at the time of sacrifice for an extreme limiting dilution assay with $1 \times 10^3$–$5 \times 10^4$ mCherry-H2B PANCA6L Gluc-CC and Gal-CC. **l** Flow cytometric quantification of mean of percentage ± sd of double-positive mCherry+CD133+ cells after digestion of Gluc-CC and Gal-CC tumors. Number ($n$) of samples analyzed are indicated. (Holm–Sidak $t$ test statistical analysis). *$p = 0.044$; ns = 0.066; **$p = 0.0022$. **m** Number of tumors obtained, the frequency (Freq), and percentage (%) of CSCs present in tumors as a function of the dilutions tested.

---

validated by liquid chromatography–mass spectrometry- or nuclear magnetic resonance-based metabolomics, we indirectly confirmed the latter by enzyme-linked immunosorbent assay (ELISA) analysis of lactate production (Supplementary Fig. 3d). At the level of total ATP, no significant differences were observed between Gluc-CC and Gal-CC at the time point examined (Supplementary Fig. 3e), which reflects differences in the kinetics of ATP production and/or consumption or glycolytic rate compensations between Gluc-CC and Gal-CC[18].

Transcription of genes related to mitochondrial activity and electron transport chain complexes (NDUFA9-Complex I and MT-COI-Complex IV) and mitochondrial 16S ribosomal RNA gene (MT-RNR2) also significantly increased in Gal-CC (Fig. 3e). Importantly PGC1α, a key regulator of energy metabolism that stimulates mitochondrial biogenesis, promotes OXPHOS metabolism, and is overexpressed in PaCSCs[8], was transcribed to significantly higher levels in Gal-CC. Moreover, NRF2 and PARKIN, related to the regulation of antioxidant proteins[19] and recycling of mitochondria via the coordinated process of mito-fusion, mito-fission, and mitophagy[20], were also significantly increased (Fig. 3e). Analysis of MT-RNR1 DNA levels revealed increased mtDNA copy numbers in Gal-CC compared to Gluc-CC (Fig. 3f), consistent with more mitochondrial biogenesis. In line with the latter, we observed a significant increase in the percentage of MTDR and MT-CMXRos fluorescent area in Gal-CC compared to Gluc-CC (Fig. 3g–i). Interestingly, we could observe that the cells on the borders of the clones showed more intense staining (Supplementary Fig. 3g), which could suggest a greater energy demand, perhaps necessary for a higher duplication rate or migration[21,22]. The mitochondria of Gal-CC were visually denser (Fig. 3g), and immunofluorescence (IF) analysis revealed a significant increase in TOM20 staining (~2-fold) in Gal-CC (Fig. 3j, k), which was also used to show that the mitochondrial network in Gal-CC was more extensive compared to Gluc-CC (Supplementary Fig. 3f). In addition, we observed greater expression of PARKIN by flow cytometric analysis (Supplementary Fig. 3h, i) and by IF (Fig. 3j, l) in Gal-CC, indicating increased mitochondrial recycling.

**Cell cycle and quiescence in Gal-CC.** To dissect Gal-CC and Glu-CC at the transcriptional level, we performed RNAseq analysis to identify differentially regulated pathways. In line with an enrichment in PaCSCs, numerous CSC-associated pathways were enriched in Gal-CC, such as mammalian target of rapamycin C1 (mTORC1), hypoxia, phosphoinositide-3 kinase/AKT/mTOR, interleukin-6 (IL-6)/Janus-activated kinase/signal transducer and

activator of transcription factor 3 (STAT3), IL-2/STA5, and Wnt-β-catenin signaling (Fig. 4a and Supplementary Fig. 4a). In addition, and as a validation our OCR results and metabolic analyses, OXPHOS was significantly enriched in Gal-CC (Fig. 4a and Supplementary 4a). Finally, we observed a significant reduction in G2/M checkpoint genes and E2F targets (Fig. 4a and Supplementary 4b), including CDK1, which controls entry into mitosis, the phosphatase CDC25A that controls entry into and progression through mitosis and S phase, and principal regulators of the mitotic checkpoint, including MAD2L1 and BUB1B[23,24].

Based on these RNAseq data and our IncuCyte Zoom System analysis (Fig. 1a, b), where we observed less proliferation in Gal-CC, we performed a propidium iodide (PI)-based cell cycle analysis. As expected, we measured a higher percentage of cells in G0/G1 and S phases in Gal-CC, whereas more cells were in G2/M phase in Gluc-CC (Fig. 4b), which is in line with cell cycle inhibition and the idea that CSCs have a slower cell cycle, resembling a quiescent-like state, compared to non-CSCs. To discard whether senescence was affecting cell proliferation, β-galactosidase activity was assessed (Fig. 4c), but the percentage of β-galactosidase-positive area was significantly lower (twofold) in Gal-CC compared to Gluc-CC (Fig. 4d), suggesting that Gal-CC are less senescent. For this reason, we performed a quiescence assay by labeling cultures with the red fluorescent plasma membrane lipophilic dye PKH26 (Fig. 4e) to evaluate proliferation at different times (Fig. 4f). The percentage of PKH26+ cells were very similar at 3 days (Fig. 4g–j); however, at 7 days a significant reduction of >15% in the percentage of PKH26+ Gluc-CC was observed, while the percentage of PKH26+ Gal-CC did not significantly change (Fig. 4g–j). Finally, at 14 days, a reduction of >50% in the percentage of PKH26+ Gluc-CC was observed, whereas in Gal-CC the percentage of PKH26+ cells was reduced only by 10–15% (Fig. 4g–j). To determine whether these cultures were permanently fixed in a quiescent-like state, galactose-containing medium was substituted for glucose-containing medium, and in as little as 7 days, all of the Gal-CC showed a significant reduction in the percentage of PKH26+ cells (~80%) (Fig. 4k), indicating a reversal in the observed quiescent-like state. Of note, this decrease was greater/faster than that observed in Gluc-CC at 7 days (Fig. 4g–j).

**Autophagy increases in Gal-CC.** CSCs use autophagy to regulate quiescence and self-renewal and have more autophagic flux to help them better survive in stressful conditions (e.g., lack of nutrients, hypoxia, and chemotherapeutics)[25,26]. By brightfield microscopy, morphological differences between Gluc-CC and

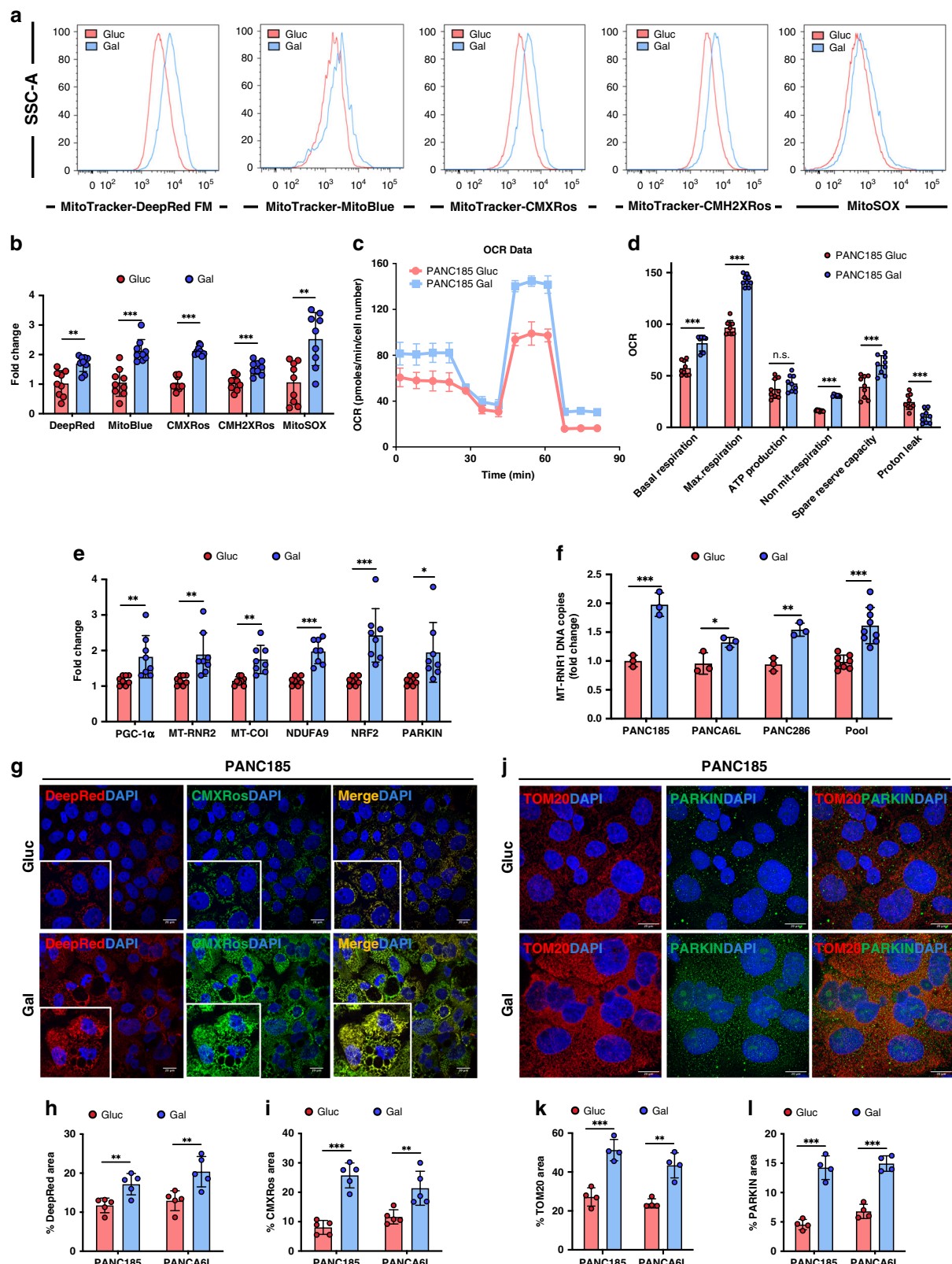

Gal-CC were observed (Fig. 5a), with Gal-CC forming smaller and more compact clones with significantly more vacuolar bodies in their cytoplasm than Gluc-CC (Fig. 5a). TEM analysis confirmed enrichment in electron-dense cellular structures, many reminiscent of double membranes/multi-lammelar autophagic vesicles[27] in Gal-CC (Fig. 5b). Quantification of autophagic

vesicles revealed greater numbers in Gal-CC compared to Gluc-CC (Fig. 5b, c). IF analysis of LC3B I–II and LAMP-1, both essential proteins related to the autophagic process, revealed that Gal-CC had a higher percentage of LC3B I–II and LAMP-1 fluorescent area compared to Gluc-CC (Fig. 5d, f), indicating both increased autophagy and lysosome numbers. To confirm these

**Fig. 3 PDAC cells increase OXPHOS in the presence of galactose. a** Representative flow cytometric histograms of the mean fluorescence intensity of cells expressing the indicated markers. **b** Quantification of mean fold change ± sd of the different markers in **a** ($n = 9$ biological replicates; Holm–Sidak $t$ test statistical analysis). Gluc was set as 1.0. **$p = 0.00107$; ***$p = 0.00007$; ***$p < 0.000001$; ***$p = 0.00027$; **$p = 0.0013$. **c** Representative plot showing mean ± sd of oxygen consumption rate (OCR) at the indicated times, normalized to cell number, which were treated with distinct inhibitors of mitochondrial function: O (oligomycin), F (FCCP), A (antimycinA), and R (rotenone). Continuous OCR values (pmoles/min/cell) are shown. **d** Measured and calculated mean ± sd OCR parameters ($n = 3$ biological replicates with 3 readings; Holm–Sidak $t$ test statistical analysis). ***$p = 0.000003$; ***$p < 0.000001$; ns = 0.223; ***$p < 0.000001$; ***$p = 0.0003$; ***$p = 0.0005$. **e** RT-qPCR analysis of relative mRNA expression of different genes related to mitochondrial activity and electron transport chain complexes and ROS neutralization. mRNA levels are normalized to β-actin ($n = 8$ biological replicates; Holm–Sidak $t$ test statistical analysis). Data are presented as fold change ± sd. Gluc was set as 1.0. **$p = 0.0085$; **$p = 0.055$; **$p = 0.00103$; ***$p = 0.00032$; *$p = 0.018$. **f** RT-qPCR analysis of mean relative mitochondrial 12S ribosomal RNA gene (MT-RNR1) levels for each PDAC culture and pooled samples. Data are normalized to β-actin ($n = 3$ biological replicate; Holm–Sidak $t$ test statistical analysis). Data are presented as fold change ± sd. Gluc was set as 1.0. **$p = 0.0017$; *$p = 0.03$; **$p = 0.0029$; ***$p = 0.000028$. **g** Representative fluorescent images of Mitotracker DeepRed FM (red), CMXRos (green), and DAPI (blue) staining. **h, i** Quantification of mean percentage of fluorescent area ± sd for **h** MitoTracker-DeepRed FM mitochondrial biomarker or the **i** CMXRos mitochondrial membrane potential marker, using ImageJ. ($n = 5$ independent fields; Holm–Sidak $t$ test statistical analysis). For **h**: **$p = 0.0064$; **$p = 0.0073$. For **i**: ***$p = 0.000039$; **$p = 0.0083$. **j** Representative immunofluorescent images of TOM20 (red), PARKIN (green), and DAPI (blue) staining. **k, l** Quantification of mean percentage of fluorescent area ± sd for **k** TOM20 mitochondrial marker or **l** PARKIN, using ImageJ ($n = 4$ independent fields; Holm–Sidak $t$ test statistical analysis). For **k**: ***$p = 0.00054$; **$p = 0.0013$. For **l**: ***$p = 0.00012$; ***$p = 0.000099$.

results, we analyzed the basal levels of LC3B I and LC3B II by western blotting (WB) analysis (Fig. 5g) and observed that the levels of LC3B II increased in each PDAC culture in the presence of galactose (Fig. 5h), as well as the expression of *ATG5*, a key regulator of autophagy (Fig. 5i). To definitively confirm increased autophagy in Gal-CC, we evaluated autophagic flux. After treatment with Bafilomycin A1, a known inhibitor of the late phase of autophagy, the levels of LC3B I and LC3B II were evaluated by WB analysis (Fig. 5j), and a significant increase in the autophagic flux was observed for Gal-CC (Fig. 5k).

**Chemoresistance is enhanced in Gal-CC.** Chemoresistance is an inherent characteristic of PaCSCs, it is linked to increased autophagy[25,26], and overexpression of ATP-binding transporters (ABC transporters), such as ABCG2, are known effectors of chemoresistance across multiple tumor types[6]. In Gal-CC, ABCG2 transcript levels significantly increased ~2-fold compared to Gluc-CC (Supplementary Fig. 5a). In addition, we evaluated the expression of genes related to the transport of Gemcitabine (GEM), one of the frequently used chemotherapeutics in the treatment of PDAC, specifically the GEM efflux transporters hENT1 and hENT2[28] and the GEM influx transporter CNT1[29]. CNT1 was significantly lower in Gal-CC compared to Gluc-CC, whereas hENT1 and hENT2 were significantly higher (Supplementary Fig. 5b). In view of these results, we next evaluated the capacity of relapse/survival of the different Gluc-CC and Gal-CC after treatment with different chemotherapeutics (Mitoxantrone (MTX), Doxorubicin (DXR), and GEM; Supplementary Fig. 5c) and observed significantly less cell death for all treatments and across all PDAC cultures in the presence of galactose compared to glucose (Supplementary Fig. 5d–f). While resistant to chemotherapeutics, Gal-CC were highly sensitive to OXPHOS metabolic drugs, such as Menadione, Rotenone, Resveratrol, and Metformin, but not to the glycolytic irreversible inhibitor, 2-deoxy-glucose (2-DG; Supplementary Fig. 5g–k).

**Immune evasion properties are enriched in Gal-CC.** Many studies have also shown that there is a clear relationship between quiescence and immune evasion mechanisms and metastasis[4,30]. Since little is known about PaCSC immune evasion, we evaluated the expression of ligands and receptors related to immune suppression of T cells (programmed death-ligand 1 (PD-L1)); antiphagocytic function (CD47); and invasion, migration, and metastasis (CD155 and CD206) and observed that all were significantly upregulated in Gal-CC compared to Gluc-CC (Fig. 6a,

b). Interestingly, ULBP2/5/6 ligands related to the activation of natural killer (NK) cell receptors were significantly down-regulated in Gal-CC (Fig. 6a, b).

At the transcriptional level, we evaluated different epithelial–mesenchymal transition (EMT)-related genes. *ZEB1*, a key regulator of invasion and metastasis[31], as well as the EMT transcription factors (TF) *SNAI2* and *SNAI1* were significantly higher in Gal-CC compared to Gluc-CC, whereas *CDH1* and *VIM* were unchanged (Fig. 6c). We next evaluated transforming growth factor-β (TGFβ) secretion levels, one of the most important cytokines implicated in stemness, EMT, and extracellular matrix regulation, and measured significantly higher levels in Gal-CC compared to Gluc-CC (Fig. 6d). We also found a significant overexpression of nuclear factor-κB in Gal-CC, an important TF that mediates cytokine secretion[32] (Fig. 6e). Based on this result, we performed a cytokine array. Overall, we found differences in the levels of expression of different secreted cytokines or inflammatory-associated molecules in both culture conditions (Fig. 6f and Supplementary Fig. 6a, b). Significant differences were found in C-C motif chemokine ligand 2 (CCL2), C-X-C motif chemokine ligand 12 (CXCL12), granulocyte macrophages colony-stimulating factor (GM-CSF), and soluble intercellular adhesion molecule (ICAM)-1 (Fig. 6g), as well as CCL5, CXCL1, Serpin E1, and IL-8 (Fig. 6h). Interestingly, the factors that significantly increased in Gal-CC have been associated with higher stemness (CCL2, IL-8); tumorigenesis (CCL2, CXCL12, IL-8); chemoresistance (CXCL12); metastasis, migration, and invasion (CXCL12, IL-8); and quiescence (CXCL12). On the other hand, the decrease of the other factors has been correlated with lower chemotaxis, communication and activation of immune cells like T cells, basophils, neutrophils (ICAM-1, CCL5, CXCL1), higher quiescence and infiltration of pro-inflammatory macrophages (GM-CSF), higher metastasis, and stemness (Serpin E1)[33].

To test whether Gal-CC are more immunoevasive, we performed functional in vitro immune evasion assays. Co-culturing DilC-stained (red) macrophages from 3 healthy donors with PKH67-stained (green) tumor cells (Glu- and Gal-CC) (Fig. 6i) resulted in less double-positive DilC+PKH67+ cells (indicative of macrophage-mediated phagocytosis) (Fig. 6j) for Gal-CC compared to Gluc-CC (Fig. 6k), indicating that macrophages are less able to detect and phagocytose Gal-CC than Gluc-CC, likely due to the overexpression of the "don't eat me" CD47 receptor[34] on the surface of Gal-CC (Fig. 6a, b). Next, activated T cells were co-cultured with tumor cells (Glu- and Gal-CC). Regardless of the donor and PDAC cell line used, T cells were less efficient at eliminating Gal-CC

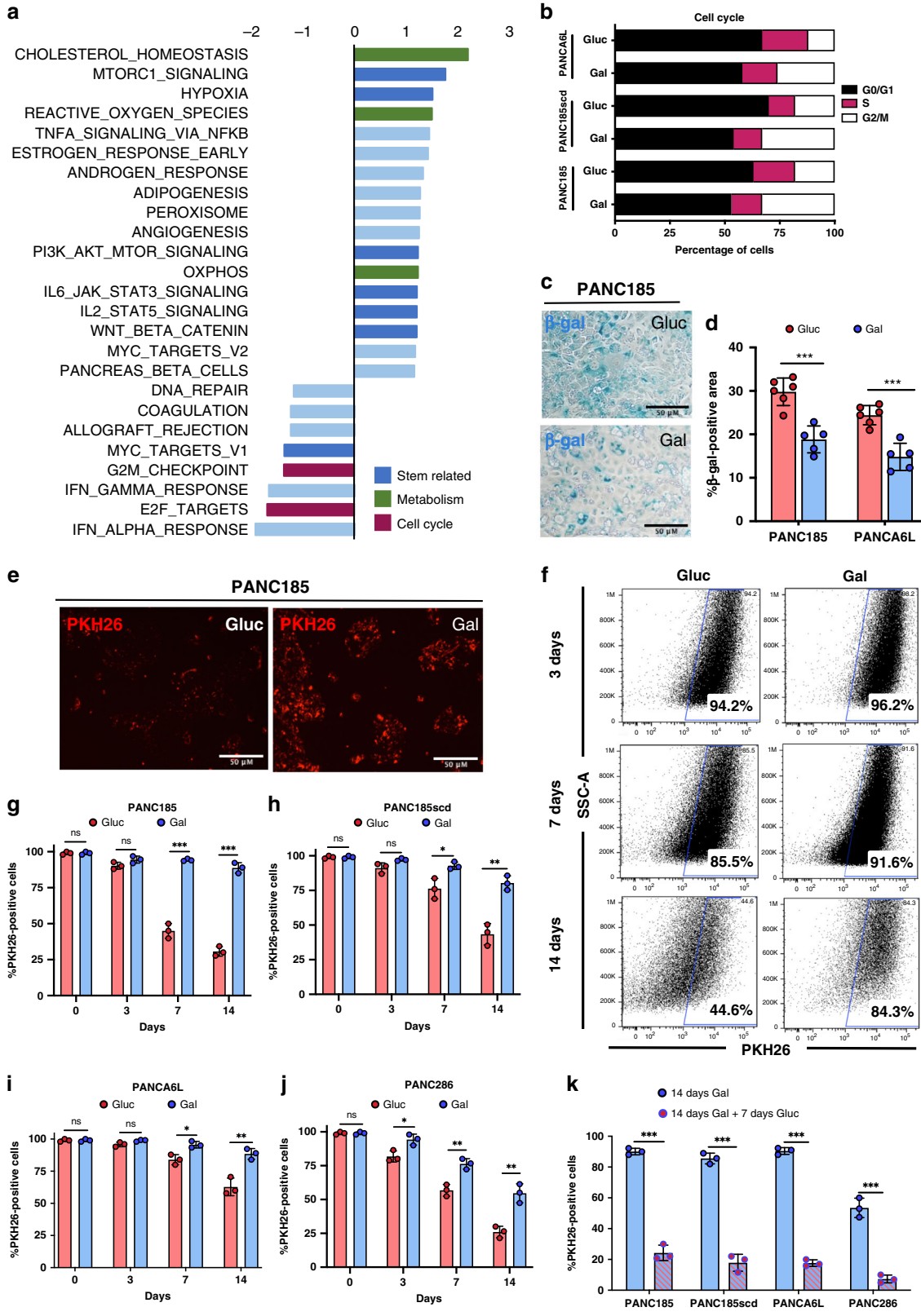

compared to Gluc-CC when they were co-cultivated (Fig. 6l, n). Importantly, the levels of cell death detected in T cell/Gal-CC co-cultures was very similar to control cultures (without T cells) for Donor 1 (Fig. 6l) and Donor 3 (Fig. 6n). T cells from Donor 2, in general, displayed higher cytolytic capacity (Fig. 6m).

**Upregulated in vivo immune evasion and metastasis in Gal-CC.** To functionally test the invasive capacity of Gal-CC and Gluc-CC, zebrafish and NOD-SCID mouse models of cellular invasion were employed. Zebrafish embryos were microinjected into circulation (Duct of Cuvier), as previously described[35]. While at 1 day post-

**Fig. 4 CSC-specific pathways, including quiescence, are enriched in Gal-CC. a** Gene sets enriched in the transcriptional profile of PANCA6L Gal-CC compared with Gluc-CC. Shown are the NES (normalized enrichment score) values for each pathway using the Hallmark genesets, with nominal $p$ value of <0.05 and FDR < 25%. Stem-related pathways, metabolism pathways, and cell cycle pathways are color coded. **b** Cell cycle analysis of Gluc-CC and Gal-CC PDAC cultures (PANC185, PANC185scd, and PANCA6L) determined by flow cytometric analysis using propidium iodide. **c** Representative brightfield images by β-galactosidase staining of PANC185 Gluc-CC and Gal-CC (blue). Scale bar = 50 μM. **d** Quantification of mean percentage of positive area ± sd of β-galactosidase in Gluc-CC and Gal-CC, using ImageJ ($n = 6$ independent fields; Holm–Sidak $t$ test statistical analysis). ***$p = 0.00027$; ***$p = 0.00021$. **e** Representative fluorescent images of PANC185 Gluc-CC and Gal-CC stained with PKH26 (red) at 14 days. Scale bar = 50 μM. **f** Representative flow cytometric dot plots of the percentage of PKH26-positive cells in Gluc-CC and Gal-CC at different times (3, 7, and 14 days). **g–j** Quantification of mean percentage ± sd of PKH26-positive cells at the indicated times (0, 3, 7, and 14 days) for Gluc-CC to Gal-CC ($n = 3$ biological replicates per culture; Holm–Sidak $t$ test statistical analysis). For **g** PANC185: ns > 0.99; ns = 0.09; ***$p = 0.000078$; ***$p = 0.000033$, for **h** PANC185scd: ns > 0.99; ns = 0.052; *$p = 0.022$; **$p = 0.0023$, for **i** PANCA6L: ns > 0.99; ns = 0.053; *$p = 0.013$; **$p = 0.0052$, and for **j** PANC286: ns > 0.99; *$p = 0.019$; **$p = 0.0042$; **$p = 0.0038$. **k** Quantification of mean percentage ± sd of PKH26-positive cells in Gal-CC after 14 days in galactose-containing media and 7 days after substitution with glucose ($n = 3$ biological replicates; Holm–Sidak $t$ test statistical analysis). ***$p = 0.00003$; ***$p = 0.000057$; ***$p = 0.000002$; ***$p = 0.0003$.

injection (dpi) PANCA6L mCherry-H2B Gluc-CC migrated more efficiently to the tail compared to Gal-CC (Fig. 7a, b), Gluc-CC were less viable at later time points, as evident by the loss in fluorescence observed at 4 and 6 dpi (Fig. 7c, d). For PANC185scd, similar results were observed (Supplementary Fig. 6c–f). To extend these observations, NOD-SCID mice were tail vein injected with Gluc-CC or Gal-CC and their capacity to disseminate and invade to distant organs was assessed at 10 dpi (short time) and 3 months pi (long time) (Fig. 7e).

At 10 dpi, while there was no significant increase in mCherry-positive cells in the spleen, liver, or lungs, there was a significant increase in the pancreas and blood in mice injected with Gal-CC (Fig. 7f and Supplementary Fig. 7a). However, at 3 months pi, we found that mice injected with Gal-CC presented with significantly higher percentages of mCherry-positive cells in most organs compared to mice injected with Gluc-CC, including the liver, one of the main metastatic sites in PDAC patients (Fig. 7g and Supplementary Fig. 7a). In some organs, such as kidneys, thymus, and lymphatic nodes, no significant differences were observed (Supplementary Fig. 7c). Results were confirmed and validated ex vivo using IVIS imaging (Fig. 7h, i) or by immunohistochemical (IHC) analysis of human cytokeratin-19 (Supplementary Fig. 7a, b). Interestingly, we found numerous and abnormally large lymphatic nodes in mice injected with Gal-CC, which were not detected in the mice injected with Gluc-CC (Fig. 7h, yellow squares). Lastly, we analyzed in each organ the percentage of tumor-associated macrophages (TAMs) (CD45$^+$CD11b$^+$F4-80$^+$), which are known to play an important role both in the primary tumor and for pre-metastatic niche formation. In mice injected with Gal-CC, we observed at 10 dpi significantly lower TAMs in the liver and pancreas (Fig. 7j); however, at 3 months pi the population of TAMs increased significantly in all the organs analyzed except the lungs and kidneys (Fig. 7k and Supplementary Fig. 7d).

## Discussion

We have developed a 2D in vitro system for long-term and sustained enrichment of PaCSCs based on forced OXPHOS. By culturing cells with galactose in lieu of glucose, cells that are exclusively dependent on glycolysis (non-PaCSCs) cannot survive due to differences in ATP yield and availability of glycolysis intermediates. Indeed, we observed cell death in the presence of galactose that coincided with an increase in the PaCSC population, confirming that non-PaCSCs cannot survive in galactose. To confirm the latter, we placed purified non-CSCs in galactose and observed that, while galactose favors the conversion of non-PaCSC in PaCSCs (i.e., plasticity), this contribution is secondary compared to the enrichment achieved due to loss of the non-PaCSC compartment. While we have not defined the mechanism

of non-PaCSC plasticity, it is plausible that non-PaCSCs could increase their mitochondrial mass to survive in metabolically restrictive conditions.

In our OXPHOS-promoting system, we observed a profound increase in traditional CSC features, such as the expression of multiple CSC biomarkers, overexpression of pluripotency-associated genes and stem-related pathways (validated by RNA-seq), increased NANOG promoter activity, enhanced self-renewal capacity, and, most notably, significantly higher in vivo tumor-igenicity. Other studies using alternate carbon sources (i.e., fructose) in breast cancer cells observed similar results but their studies were not as exhaustive and did not evaluate all of the phenotypes presented herein[36]. Not surprisingly, we did not obtain 100% CSC biomarker staining in Gal-CC, which we reconcile by acknowledging that heterogeneity exists within the CSC population(s), making it impossible to detect all CSC clones as not all CSCs (or OXPHOS-dependent hybrid CSCs) will express the same single biomarker or combinations of different biomarkers. Nonetheless, the increase observed in defined CSC markers (>15%) and others (mitochondrial markers of approximately 100% and immune evasion markers of >50%) clearly indicate an enrichment comparable to that obtained with traditional approaches[6,7,13]. Apart from traditional CSC features, we also observed that Gal-CC were more slow cycling than Gluc-CC, linking OXPHOS to a quiescent-like state, a feature typically associated with a CSC's survival and metastatic capacity[4]. Importantly, the latter was not permanent and/or irreversible, as Gal-CC are highly plastic and can exit this slow-cycling state in accordance to the carbon source available. Whether OXPHOS metabolism is favored by quiescent CSCs as a means of maintaining mitochondrial membrane potential, ATP production, and mtDNA content is still to be determined[21].

Many studies have shown that slow-cycling/dormant tumor cells are also more chemoresistant, have epithelial–mesenchymal plasticity (EMP), and are better able to evade the immune system, facilitating post-treatment relapse, tumor cell dissemination, and survival in hostile environments[37,38]. Along these lines, Gal-CC were more chemoresistant to a panel of traditional chemotherapeutics, largely due to the overexpression of the ABCG2 transporter and the different levels of expression of genes related to GEM transport. In contrast, and as expected, Gal-CC were more sensitive to OXPHOS metabolic drugs and less sensitive to the glycolytic irreversible inhibitor 2-DG. Moreover, we also observed increased autophagy in Gal-CC, suggesting that autophagy is also likely an important mechanism used by OXPHOS-dependent cells in metabolically restrictive conditions, such as those faced by CSCs in the tumor microenvironment (TME) where oxygen and nutrients are reduced. It has also been observed that autophagy is strongly linked to quiescent/dormant processes in cancer

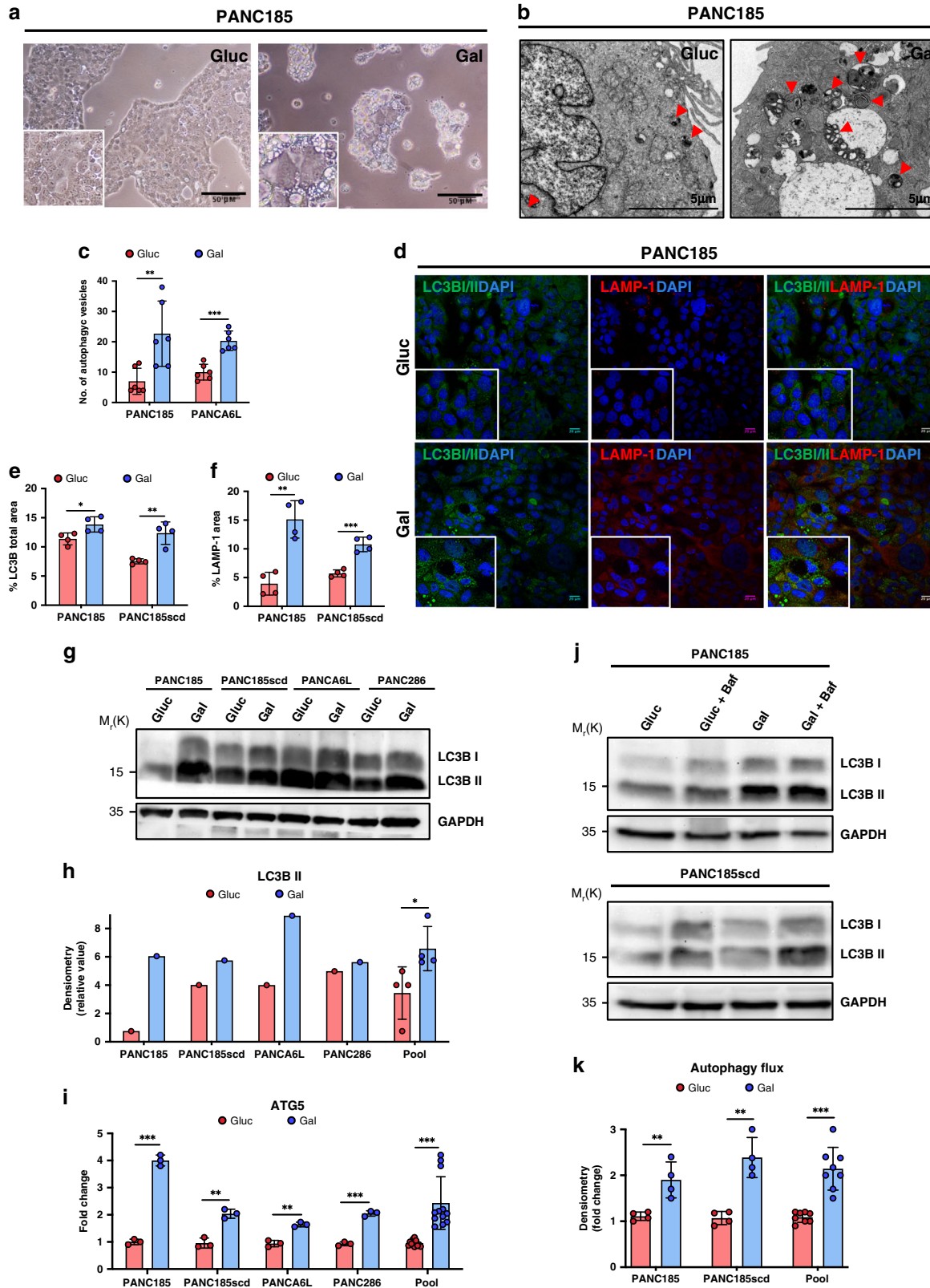

cells[39,40] as well as EMT and EMP. For example, two recent reports demonstrate that autophagy is intimately linked to the capacity of cancer cells to transition between epithelial and mesenchymal states[38,41]. EMP is now considered a critical process for tumor cell dissemination, metastatic colonization, and chemoresistance. Not only were Gal-CC more chemoresistant but

also different EMT TFs were significantly overexpressed, including *ZEB1*, which was recently linked to PDAC metabolic plasticity in vivo[42]. In fact, biological associations between EMT and metabolic reprogramming in cancer have been described and shown to be a consequence of EMT TF signaling[43]. Thus the overexpression of *ZEB1* in cultures enriched in OXPHOS-

**Fig. 5 Autophagy increases in Gal-CC. a** Representative brightfield images of 14-day PANC185 Gluc-CC or Gal-CC, taken by optical microscopy. Scale bar = 50 μM. **b** Representative transmission electron micrographs of 14-day PANC185 Gluc-CC or Gal-CC. Autophagic structures (red arrow heads). **c** Mean number of autophagic vesicles ± sd in the TEM images of 14-day PANC185 and PANCA6L Gluc-CC and Gal-CC, quantified by ImageJ ($n = 6$ independent fields; Holm–Sidak $t$ test statistical analysis). \*\*$p = 0.0078$; \*\*\*$p = 0.00011$. **d** Representative immunofluorescence images of PANC185 Gluc-CC and Gal-CC stained with anti-LC3BI/II (green), anti-LAMP-1 (red), and DAPI (blue). **e** Mean percentage of fluorescent area ± sd for LC3BI/II comparing Gluc-CC and Gal-CC with ImageJ ($n = 4$ independent fields; Holm–Sidak $t$ test statistical analysis). \*$p = 0.024$; \*\*$p = 0.0027$. **f** Mean percentage of fluorescent area ± sd for LAMP-1 in PANC185 and PANCA6L Gluc-CC and Gal-CC, using ImageJ ($n = 4$ independent fields; Holm–Sidak $t$ test statistical analysis). \*\*$p = 0.00107$; \*\*\*$p = 0.00038$. **g** WB analysis of LC3BI/II in Gluc-CC and Gal-CC PDAC cultures. GAPDH was used as loading control. **h** Densitometric quantification of LC3BII expression in **g**, normalized to GAPDH levels and expressed as relative values for each tumor or pooled samples from the same immunoblot ($n = 4$ biological replicates in Pool; Holm–Sidak $t$ test statistical analysis). \*$p = 0.041$. **i** RT-qPCR analysis of relative mRNA expression levels of *ATG5* in Gluc-CC and Gal-CC PDAC cultures. mRNA expression levels for each target gene are normalized to β-actin levels ($n = 3$ biological replicates per tumor, $n = 12$ for pool; Holm–Sidak $t$ test statistical analysis). Data are presented as mean fold change ± sd. Gluc was set as 1.0. \*\*\*$p = 0.00002$; \*\*$p = 0.0016$; \*\*$p = 0.00105$; \*\*\*$p = 0.000088$; \*\*\*$p = 0.000029$. **j** WB analysis of LC3BI/II in Gluc-CC and Gal-CC untreated (Gluc, Gal) or treated with the autophagy inhibitor (Baf = Bafilomycin). GAPDH was used as loading control. **k** Mean fold change ± sd of densitometric quantification of autophagic flux (LC3B-II accumulation after autophagy inhibitor addition, subtracting their respective control expression levels), ($n = 4$ independent immunoblots for each culture; $n = 8$ pooled values, Holm–Sidak $t$ test statistical analysis). Gluc was set as 1.0. \*\*$p = 0.0077$; \*\*$p = 0.0012$; \*\*\*$p = 0.000022$.

dependent CSCs is not surprising but does validate the system and its biological relevance.

Finally, we show that OXPHOS-driven PaCSCs exhibit immune evasive phenotypic and functional properties. Indeed, studies have speculated that CSCs may maintain only limited antigen-presentation levels and hence are invisible to the immune system, but it is also possible that CSCs possess antigen editing-independent capacities that may endow them with immunoevasive properties[5,33,44]. In this sense, Gal-CC secreted more TGFβ and acquired a previously uncharacterized immune evasive molecular signature, enabling them to evade immune innate cells (i.e., macrophages and NK cells) and immune adaptive cells (T cells) thanks to the modulation of different immune-associated receptors, ligands, and secreted cytokines. As a consequence, Gal-CC were more invasive and metastatic in vivo and were able to modulate the immune infiltrating cell profile at early and late times post-injection, attracting fewer immune cells during the early stages of extravasation, while at later times promoting the accumulation of pro-tumor immune cells, specifically TAMs, which facilitate establishment of the niche and TME.

It is not surprising that mitochondria are important mediators of tumorigenesis[21]. While we have previously shown that PaCSCs use OXPHOS to meet their energy requirements[8], herein we expand upon this observation by showing that Gal-CC have a denser and more extensive mitochondria network and have more functional mitochondria compared to glycolytic cells (non-PaCSCs). Indeed, Gal-CC overexpress mitochondrial-associated genes, such as *PGC1α*, which promotes mitochondrial biogenesis and OXPHOS and these processes are linked to cancer cell migration, EMP, and immune evasion[21]. Likewise, it has been shown in other systems that the aberrant expression of immune evasion receptors (e.g., PD-L1) is metabolically controlled[45], and in cancer, correlations between metabolic alterations, EMT, and PD-L1 upregulation have been demonstrated[46]. Thus our system has allowed us to rigorously demonstrate that the metabolism of PaCSCs is intimately linked to their stem, slow-cycling, meta-static, and immunoevasive capacities.

## Methods

**Primary human PDAC cells**. PDAC PDXs were obtained from Dr. Manuel Hidalgo under a Material Transfer Agreement with the Spanish National Cancer Centre (CNIO), Madrid, Spain (Reference no. I409181220BSMH) and were originally described and genetically characterized in ref. [47]. For primary cultures, PDX-derived tumor tissue fragments were minced, enzymatically digested with collagenase (Cat no.

07416, Stem Cell Technologies) for 60 min at 37 °C, and after centrifugation for 5 min at $500 \times g$ the pellets were resuspended and cultured in RPMI, 10% fetal bovine serum (FBS), and 50 units/ml penicillin/streptomycin as described previously[7]. PDAC PDX-derived cultures are referred to by a random number designation [i.e., PANC185, PANC185scd (single-cell-derived)[7], PANCA6L, and PANC286]. Primary cultures were tested for Mycoplasma at least every 4 weeks.

**Establishment of OXPHOS-competent and -independent cultures**. Low passage (<15) PDX-derived primary PDAC cultures were trypsinized and seeded at a concentration of 800,000 cells in p100 plates with RPMI medium supplemented with 10% FBS and 50 units/ml of penicillin and streptomycin. After 24 h, cells were cultured with either glucose-free Dulbecco's Modified Eagle Medium (DMEM; Thermo Fisher Scientific) supplemented with 5 mM glucose (0.9 g/l), 10% FBS, 50 units/ml of penicillin and streptomycin, and 1 mM of pyruvate [Glucose: OXPHOS-independent conditions] or glucose-free DMEM medium (Thermo Fisher Scientific) supplemented with 5 mM galactose (0.9 g/L), 10% FBS, 50 units/ml of penicillin and streptomycin, and 1 mM of pyruvate [Galactose: OXPHOS-competent enriched conditions]. Sugar concentrations of 5 mM were chosen to mimic physiological sugar levels (glucose, 5 mM) and to avoid potential biological artifacts mediated by supraphysiological sugar levels. Media for both conditions were changed every day, over a period of 14 days, after which cells were collected for further processing and analysis, as described below. Earlier time points (e.g., 7 days) were also tested, but maximum levels of markers and gene expression were obtained at 14 days, and thus this time point was chosen for all experiments described herein. Gal-CC cultures can be maintained in culture indefinitely (>70 days tested), trypsinized, and passaged if necessary, as well as cryopreserved for future thawing and usage.

**Flow cytometry and sorting**. Cells were resuspended in Flow buffer [1× phosphate-buffered saline (PBS); 3% FBS (v/v); 3 mM EDTA (v/v)] before analysis with a 4-laser Attune NxT Acoustic Cytometer (Thermo Fisher Scientific). Cytometry data were acquired with the Invitrogen™ Attune™ NxT software, version 3.1.1, unless specified otherwise. For cell surface marker expression, refer to antibodies listed in Supplementary Table 1. For detection of PARKIN by cytometry, cells were fixed with 4% paraformaldehyde (PFA) in PBS for 10 min, washed, and permeabilized for 15 min with 1% Triton X-100. For Annexin-V staining, floating and attached cells were pooled and resuspended in 1× Annexin-V staining buffer containing Annexin-V-fluorescein isothiocyanate (FITC) diluted 1:20 (Cat no. 29001, Biotium, Freemont, CA) and incubated for 20 min at room temperature (RT) prior to flow cytometric analysis. For autofluorescent detection, cells were excited with blue laser 488 nm and selected as intersection with the filters 530/40 (FITC) and 580/30 (phycoerythrin (PE))[7]. For all assays, 2 mg/ml 4,6-diamidino-2-phenylindole (DAPI; Cat no. D9564, Sigma) was used to exclude dead cells with laser VL1. Data were analyzed with the FlowJo 9.3 software (Tree Star Inc., Ashland, OR.). For cell sorting, cells were resuspended in sorting buffer [1× PBS; 0.1% FBS (v/v); 3 mM EDTA (v/v)] and a FACS Vantage SE Flow Cytometer was used and events were analyzed by the BD FACSDiva™ software v.9.0 (BD Biosciences, San Jose, CA). Examples of gating strategies for all of the aforementioned cytometry-based analyses are presented in Supplementary Fig. 1d.

For mitochondria markers, Gluc-CC and Gal-CC were stained with Mitotracker Deep Red (Cat no. M22426), CM-H2XRos (Cat no. M7513), MitoTracker® Red CMXRos (Cat no. M7512), Mitotracker Green FM (Cat no. M7514), and NAO (Cat no. A1372) (all from Invitrogen) in the absence of FBS for 30 min at 37 °C, at

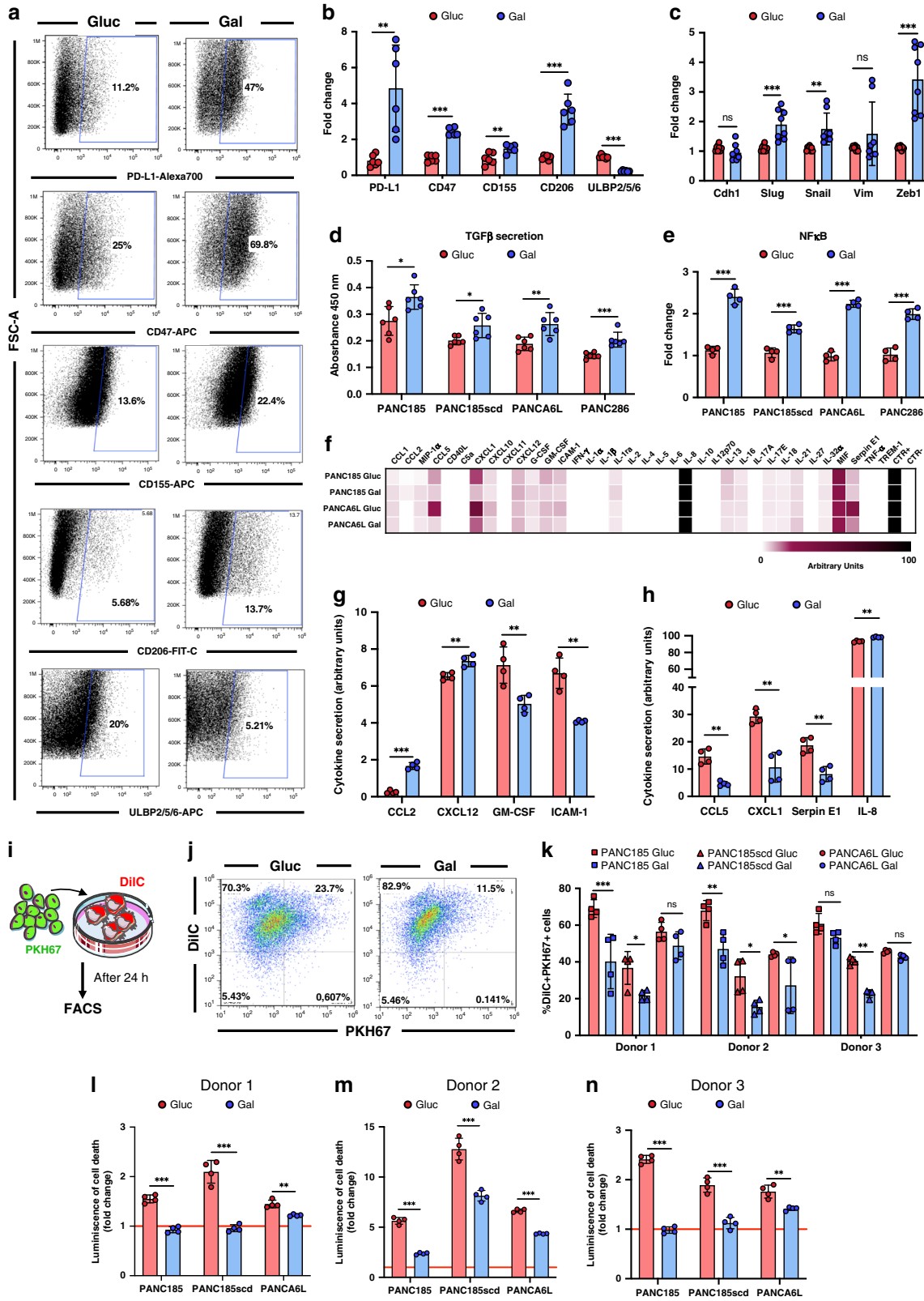

concentrations of 2, 100, and 10 nM, respectively, and 0.1 μM for Mitotracker Green FM and NAO. Fluorescence was detected using the filters RL1 (allophycoyanin), YL1 (PE), and BL1 (FITC). MitoSOX (Cat no. M36008, Invitrogen) was used at 5 μM for 30 min at 37 °C and detected with laser YL1 (PE). MitoBlue (kindly provided by Dr. JL Mascareñas, Universidad de Santiago de Compostela) was used at 10 μM for 20 min at 37 °C and detected with laser VL2. Data were analyzed with the FlowJo 9.3 software (Tree Star Inc., Ashland, OR).

**Cell cycle analysis**. One million cells were plated in 100 mm tissue-culture dishes, and after 24 h, their cell cycles were synchronized using serum-free medium. Twenty-four hours later, treatments with glucose and galactose media were initiated and cell cycle was determined 6 days later. Cells were trypsinized and fixed in cold 70% ethanol and stored at −20 °C. After 24 h, cells were incubated with 100 μg/ml RNAase A (Sigma) in PBS during 30 min at 37 °C and then stained with 50 μg/ml of PI solution (Cat no. P4864, Sigma) and stored overnight at 4 °C.

**Fig. 6 Immune evasion properties are enriched in Gal-CC. a** Representative flow cytometric plots of the percentage of cells positive for each marker. **b** Mean fold change ± sd of the different markers ($n = 6$ biological replicates; Holm–Sidak $t$ test statistical analysis). Gluc was set as 1.0. **$p = 0.0022$; ***$p < 0.000001$; **$p = 0.0058$; ***$p = 0.00002$; ***$p < 0.000001$. **c** RT-qPCR analysis of relative mRNA expression levels of different EMT-related genes. Levels were normalized to β-actin ($n = 8$ biological replicates; Holm–Sidak $t$ test statistical analysis). Data are presented as mean fold change ± sd. Gluc was set as 1.0. ns $= 0.157$; ***$p = 0.0007$; **$p = 0.0038$; ns $= 0.229$; ***$p = 0.000057$. **d** Mean absorbance levels ± sd of secreted TGFβ levels, normalized to total protein levels ($n = 6$ biological replicates for each culture; Holm–Sidak $t$ test statistical analysis). *$p = 0.0113$; *$p = 0.0164$; **$p = 0.046$; ***$p = 0.00065$. **e** RT-qPCR analysis of relative NF-κB mRNA expression levels, normalized to β-actin ($n = 4$ biological replicates for each culture; Holm–Sidak $t$ test statistical analysis). Data are presented as mean fold change ± sd. Gluc was set as 1.0. ***$p = 0.000017$; ***$p = 0.00027$; ***$p = 0.000002$; ***$p = 0.000073$. **f** Semi-quantification of the profile of cytokine secretion by pixel density of duplicated spots per sample. **g**, **h** Mean arbitrary units ± sd of secreted cytokines or inflammatory-related factors with significant changes in Gluc-CC versus Gal-CC, normalized to positive control (CTR+) ($n = 4$ biological replicates; Holm–Sidak $t$ test statistical analysis). For **g**: ***$p = 0.000053$; **$p = 0.0045$; **$p = 0.0096$; **$p = 0.0015$. For **h**: **$p = 0.0013$; **$p = 0.0017$; **$p = 0.005$; **$p = 0.0058$. **i** Representative scheme of phagocytosis assay. Schemes were constructed using unmodified images from Servier Medical Art by Servier, licensed under a Creative Commons Attribution 3.0 Unported License (https://creativecommons.org/licenses/by/3.0/legalcode). **j** Representative flow cytometric dot plots demonstrating DilC-stained macrophages and PKH67-stained Gluc-CC or Gal-CC. DilC+PKH67+ double-positive cells represent phagocytosis. **k** Mean fold change ± sd of the DilC+PKH67+ double-positive population in the phagocytosis assay using monocyte-derived macrophages from 3 different healthy donors ($n = 4$ biological replicates for each condition; two-way ANOVA statistical analysis). ***$p < 0.0001$; *$p = 0.047$; ns $= 0.651$; **$p = 0.0016$; *$p = 0.019$; *$p = 0.018$; ns $= 0.652$; ** $= 0.0095$; ns $= 0.988$. **l–n** Mean fold change ± sd in cell death after 10 days co-cultivating PDAC cells with activated T cells isolated from healthy **l** Donor 1, **m** Donor 2, or **n** Donor 3 ($n = 4$ biological replicates per condition; Holm–Sidak $t$ test statistical analysis), (red line = Control, basal toxicity set as 1.0). For **l**: ***$p = 0.000016$; ***$p = 0.00075$; **$p = 0.0013$. For **m**: ***$p = 0.000002$; ***$p = 0.00025$; ***$p < 0.000001$. For **n**: ***$p < 0.000001$; ***$p = 0.00018$; **$p = 0.0023$.

Approximately 20,000 cells/sample were analyzed using an Attune NxT Acoustic Cytometer (Thermo Fisher Scientific) with excitation at 561 nm and emission at 615 nm. The percentage of cells in each phase of the cell cycle was determined using the FlowJo 9.3 software (Tree Star Inc., Ashland, OR).

**WB analysis.** Cells were harvested in RIPA buffer (Cat no. R0278, Sigma) supplemented with protease inhibitor cocktail (Roche Applied Science, Indianapolis, IN). Fifty micrograms of protein were resolved by sodium dodecyl sulfate-polyacrylamide gel electrophoresis and transferred to polyvinylidene difluoride (PVDF) membranes (Amersham Pharmacia, Piscataway, NJ). Membranes were sequentially blocked with 1× TBS containing 5% bovine serum albumin (BSA; w/v) and 0.1% Tween20 (v/v), incubated with a 1:1000 dilution of the indicated antibodies (Supplementary Table 1) overnight at 4 °C, washed 3 times with 1× PBS containing 0.05% Tween20 (v/v), incubated with horseradish peroxidase-conjugated donkey anti-rabbit or sheep anti-mouse antibody (Amersham), and washed again to remove unbound antibody. Bound antibody complexes were detected with SuperSignal chemiluminescent substrate (Amersham) with a MyECL Imager (Thermo Fisher Scientific). Uncropped and unprocessed scans of all the main blots presented can be found in the Source data file.

**Sphere-formation assay.** PDAC spheres were generated and expanded in DMEM-F12 (Invitrogen) supplemented with B-27 (Cat no. A3582801, GIBCO) and basic fibroblast growth factor (PeproTech EC). One thousand cells/ml/well were seeded in ultra-low attachment 24-well plates (Corning) as described previously[48]. For serial passaging, spheres were harvested at day 7 using a 40-μm cell strainer, dissociated to single cells with trypsin, and then re-grown in the same conditions for 7 days (14 days total). Numbers of spheres were determined by microscopy using an inverted EVOS FL microscope (Thermo Fisher Scientific) using a ×10 objective with phase contrast.

**RNA extraction and reverse transcriptase quantitative PCR (RT-qPCR).** RNA was isolated by using the standard protocol of the guanidine thiocyanate (GTC) method[49]. One microgram of purified RNA was used for the synthesis of cDNA using the Thermo Scientific Maxima First Strand cDNA Synthesis Kit (Cat no. K1672, Thermo Fisher Scientific) according to the manufacturer's instructions, followed by RT-qPCR analysis using SYBR green and a StepOne Plus real-time thermo-cycler (Applied Biosystems). The thermal cycle consisted of an initial denaturation step of 10 min at 95 °C followed by 40 cycles of denaturation (15 s at 95 °C) and annealing (1 min at 60 °C). The results obtained for each gene were normalized with the β-actin levels. For primers used, see Supplementary Table 2.

**Lentivirus production and cell transduction.** Lentiviral particles were produced by co-transfection of 293T cells (kindly provided by Dr. Amparo Cano, Universidad Autónoma de Madrid, Spain) with the indicated plasmids using the polyethylenimine-based transfection method. Briefly, $5 \times 10^6$ 293T cells were co-transfected with 1 μg packaging plasmid psPAX2, 1 μg envelope plasmid VSVG, and 2 μg of H2B-mCherry or the NANOG reporter lentiviral vector backbone (the sequence of the construct has been previously described in the publication by Hotta et al.[50]). After 8 h, the transfection medium was changed and recombinant lentiviruses were harvested 48 and 72 h later. The virus particle-containing supernatant

was filtered through 0.45-μm PVDF membrane filters and aliquoted and stored at −80 °C until needed. For lentivirus transduction, PDAC cells were seeded in 6-well plates at a concentration of $3–5 \times 10^4$ cells/well. One milliliter of virus was directly overlaid on cells and polybrene (Cat no. 638133, Sigma) was added at a final concentration of 8 μg/ml. After 16 h, medium was changed. Stably transduced cells were obtained after mCherry-positive or YFP-positive cell sorting using a FACS Vantage SE Flow Cytometer and analyzed by the BD FACSDiva software (BD Biosciences).

**Electron microscopy.** After sorting cells for the indicated markers, cells were centrifuged and pellets were fixed with 0.1 M cacodylate buffer with a pH of 7.4 at RT and sections were processed by the UAM Electron Microscopy unit per standard protocols. Pictures were taken with a *JEM-1010* transmission electron microscope (JEOL, USA) and analyzed by Adobe PhotoShop CS4 EXTENDED V11.0 (Adobe Systems, USA).

**Optical microscopy.** All the samples processed for optical microscopy were visualized and photographed with an Axiovert 135 TV microscope (ZEISS, Germany) equipped with an Olympus DP50 digital camera and a fluorescence lamp. The images were obtained with the program ViewFinder™ 7.1 (Better Light, USA) and processed with the program Adobe PhotoShop CS4 EXTENDED V11.0 (Adobe systems, USA).

**Real-time proliferation assay.** Fifty thousand PDAC cells were seeded in a 96-well plate. After 24 h, cells were cultured with glucose or galactose media for 14 days. Real-time analysis of proliferation and autofluorescence was performed by IncuCyte® Zoom System (ESSEN BioScience, USA) taking images every 30 min for 40 h. The results were analyzed with the IncuCyte Zoom 2015A software (ESSEN BioScience, USA).

**Fluorescence microscopy.** Non-sorted and sorted cells cultured with glucose or galactose were seeded in 24-well culture dishes (Corning) and incubated at 37 °C. The mitochondrial probes MitoTracker DeepRed and MitoTracker CMXRos were added at 2 and 10 nM, respectively, for 30 min at 37 °C (all from Invitrogen), followed by two washes with 1× PBS, and a final 5 min incubation with DAPI (2 mg/ml, Sigma). MitoTracker DeepRed was excited at 644 nm and the fluorescence emitted was detected at 665 nm (far red fluorescence) and assigned red fluorescence. MitoTracker CMXRos was excited at 579 nm and the fluorescence emitted was detected at 599 nm and assigned green fluorescence. For IF assays, cells were fixed with 4% PFA in PBS for 20 min at RT, washed with PBS, permeabilized with 1% Triton X-100 in PBS for 15 min, blocked with 1% BSA in PBS for 1 h at RT, and then incubated with specific antibodies (Supplementary Table 1) in a solution of 1% BSA in PBS. ProLong® Gold Antifade Reagent with DAPI (Cat no. P36941) was then added to mark cell nuclei. The fluorescent images were collected with a laser scanning confocal microscope Zeiss 710 and analyzed using the software Zen2009 5.5 (Oberkochen, Germany).

**In vivo assays.** For in vivo tumorigenicity assays, serial dilutions of Gluc-CC and Gal-CC resuspended in Matrigel™ (Cat no. 356234, Corning) were subcutaneously injected into 8-week-old female nude mice (Hsd:Athymic Nude-

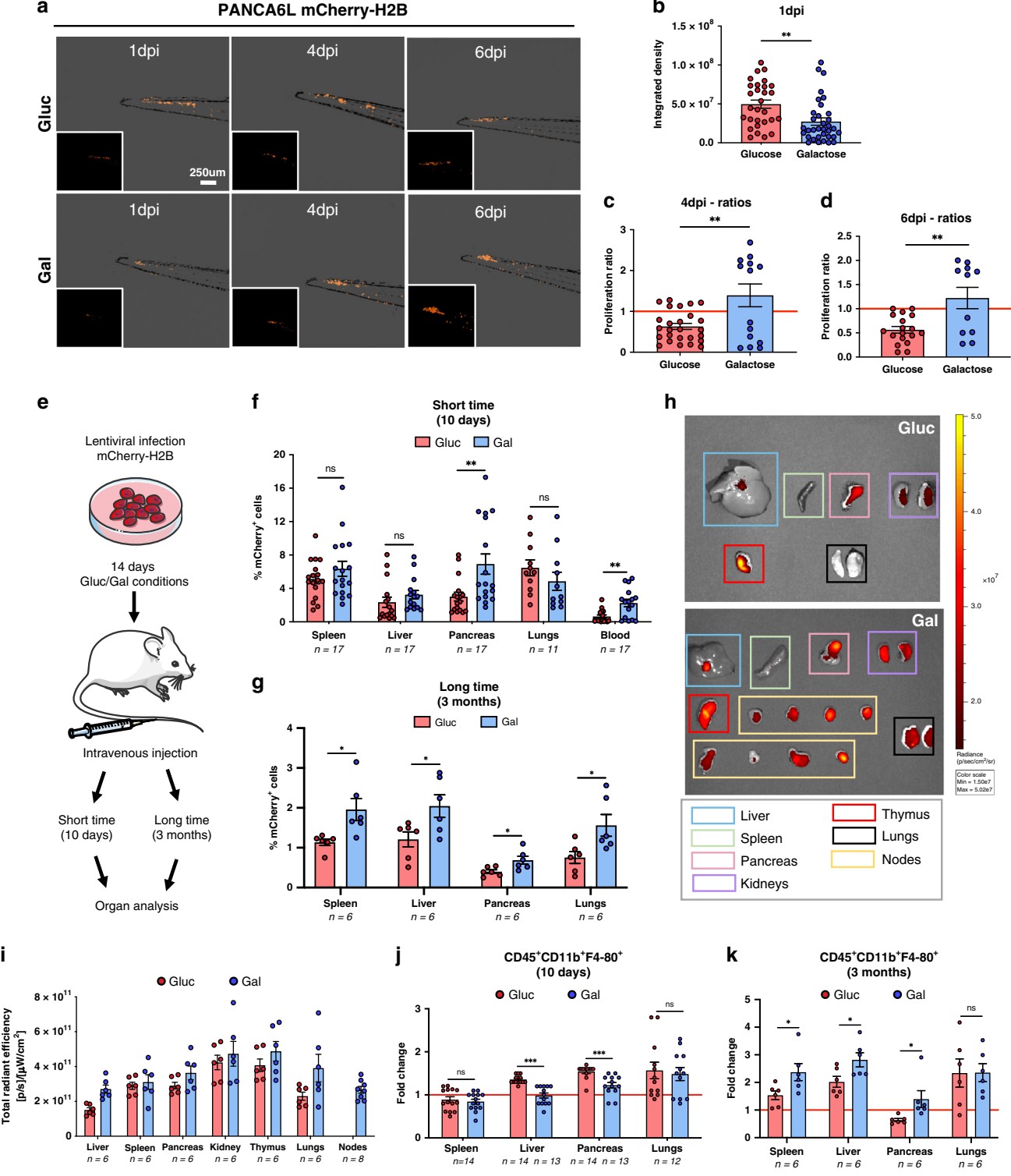

Foxn1nu/Foxn1+; Envigo) and tracked for 10–14 weeks for tumor formation. If a mouse in a specific dilution group warranted sacrifice, all of the mice (Gluc-CC and Gal-CC) in that dilution group were sacrificed in order to obtain the number of tumors for all mice at the exact same time. For the dilutions 50,000, 10,000 and 5000, mice were sacrificed at 10 weeks post inoculation. For the 1000-cell dilution group, mice were sacrificed at 14 weeks post inoculation. CSC frequencies were calculated using the ELDA software (http://bioinf.wehi.edu.au/software/elda/). For in vivo tail vein injection invasion assays, 8-week-old NOD-SCID mice (Instituto de Investigaciones Biomédicas "Alberto Sols" CSIC-UAM) were injected intravenously via the tail vein with either $5 \times 10^5$ mCherry-H2B-labeled PANC185scd or PANCA6L Gluc-CC or Gal-CC (resuspended in 0.9% physiological saline solution) using a 27-G needle. The mice were sacrificed 10 dpi or 3 months pi. Indicated

organs were collected, digested, and stained with antibodies to detect TAMs (CD45+CD11b+F4-80+) (Supplementary Table 1). In parallel, the percentage of mCherry+ cells present in each organ was determined by FACS using an Attune NxT Acoustic Cytometer (Thermo Fisher Scientific). The percentage of cells was determined using the FlowJo 9.3 software (Tree Star Inc., Ashland, OR). Organs were also analyzed by IVIS-Lumina II (Caliper Life Sciences, Hopkinton, MA, USA) and analyzed by the software Living image 3.2 (Perkin Elmer, Waltham, MA, USA).

For all in vivo experiments, mice were housed according to institutional guidelines and all experimental procedures were performed in compliance with the institutional guidelines for the welfare of experimental animals approved by the Universidad Autónoma de Madrid Ethics Committee (CEI 60-1057-A068 and CEI

**Fig. 7 In vivo immune evasion and metastasis are upregulated in Gal-CC. a** Representative images of zebrafish embryo tails at 1, 4, and 6 days post-injection (dpi). Scale bar = 250 μM. **b** Mean ± sem of the integrated density (fluorescence measured) 1 dpi (biological replicates: $n = 29$ for Glucose and $n = 36$ for Galactose; Student's $t$ test) **$p = 0.0025$. **c, d** Mean ± sem of proliferation ratio observed between Gluc-CC and Gal-CC at **c** 4 dpi and **d** 6 dpi. Proliferation ratios are represented in comparison to 1 dpi (red line) (biological replicates: 4 dpi: $n = 26$ for Glucose and $n = 14$ for Galactose; Student's $t$ test; **$p = 0.0021$; 6 dpi: $n = 18$ for Glucose and $n = 11$ for Galactose; Student's $t$ test; **$p = 0.0019$). **e** Representative in vivo invasion assay scheme. Schemes were constructed using unmodified images from Servier Medical Art by Servier, licensed under a Creative Commons Attribution 3.0 Unported License (https://creativecommons.org/licenses/by/3.0/legalcode). **f, g** Mean fold change ± sem of mCherry$^+$ cells present in different organs analyzed at **f** short time (10 days post-injection), ($n = 2$ independent experiments with a total of $n = 17$ mice; Holm–Sidak $t$ test statistical analysis) or at **g** long time (3 months post-injection). Number ($n$) of samples analyzed are indicated for each organ. ($n = 1$ independent experiment with a total of $n = 6$ mice; Holm–Sidak $t$ test statistical analysis). For **f**: ns = 0.25; ns = 0.26; **$p = 0.0058$; ns = 0.27; **$p = 0.0016$. For **g**: *$p = 0.017$; *$p = 0.034$; *$p = 0.025$; *$p = 0.027$. **h** Representative fluorescent IVIS images of different organs from mice at 3 months post-injection. **i** Mean ± sem of total radiant efficiency [p/s]/[μW/cm$^2$] of different organs from **h**. Number ($n$) of samples analyzed are indicated for each organ. ($n = 1$ independent experiment with a total of $n = 6$ mice; Holm–Sidak $t$ test statistical analysis). All data are not statistically significant. **j, k** Mean fold change ± sem of the CD45$^+$CD11b$^+$F4-80$^+$ population present in the different organs analyzed at **j** short time ($n = 2$ independent experiments with a total of $n = 17$ mice; Holm–Sidak $t$ test statistical analysis) or at **k** long time ($n = 1$ independent experiment with a total of $n = 6$ mice; Holm–Sidak $t$ test statistical analysis). Red lines indicate baseline levels of the CD45$^+$CD11b$^+$F4-80$^+$ population of control mice ($n = 6$), set as 1.0. Number ($n$) of samples analyzed for Gluc-CC- and Gal-CC-injected mice are indicated for each organ. For **j**: ns = 0.609; ***$p = 0.000003$; ***$p = 0.00057$; ns = 0.738. For **k**: *$p = 0.0361$; *$p = 0.033$; *$p = 0.0362$; ns = 0.982.

103-1958-A337) and La Comunidad de Madrid (PROEX 335/14 and PROEX 294/19) and in accordance with the guidelines for Ethical Conduct in the Care and Use of Animals as stated in The International Guiding Principles for Biomedical Research involving Animals, developed by the Council for International Organizations of Medical Sciences (CIOMS). Briefly, mice were housed according to the following guidelines: a 12-h light/12-h dark cycle, with no access during the dark cycle; temperatures of 65–75 °F (~18–23 °C) with 40–60% humidity; a standard diet with fat content ranging from 4–11%; sterilized water was accessible at all times; for handling, mice were manipulated gently and as little as possible; noises, vibrations, and odors were minimized to prevent stress and decreased breeding performance; and enrichment was always used per the facility's guidelines to help alleviate stress.

**Immunohistochemistry.** For histopathological analysis, 3-μm sections of formalin-fixed paraffin-embedded blocks were used for IHC analysis of human cytokeratin 19 using the primary and secondary antibodies and dilutions detailed in Supplementary Table 1. Antigens were visualized using 3,3-diaminobenzidine tetra-hydrochloride plus (DAB+). Counterstaining was performed with hematoxylin. Digital images were obtained using an Axiophot (Carl Zeiss) microscope with a ×10 objective and DP70 Olympus camera. Images were processed using the ImageJ v2.0.0 software (NIH, USA, http://rsbweb.nih.gov/ij/).

**Autophagy flux assay.** For autophagy flux analysis, Gluc-CC and Gal-CC were treated with the lysosomotropic reagent Bafilomycin A1 at a concentration of 150 nM (Calbiochem, Thermo Fisher Scientific, Waltham, MA, USA) for 5 h. After treatment, Gluc-CC and Gal-CC were harvested and analyzed by WB as described above. For densitometry, WB images were analyzed using PhotoShop CS4 EXTENDED V11.0 (Adobe Systems, USA). Autophagy flux compares the LC3B-II (Sigma Aldrich, St. Louis, MO, USA) levels with and without the autophagy inhibitor and is calculated as the ratio of LC3B-II in the presence and absence of the autophagy inhibitor compared to the control condition (Gal-CC versus Gluc-CC). The data were normalized to the indicated housekeeping protein.

**Quiescence assay.** Gluc-CC and Gal-CC were labeled with PKH26 red fluorescent cell membrane labeling dye (Cat no. MINI26-1KT, Sigma) according to the manufacturer's instructions. At the indicated days post staining, for a total of 14 days, cells were trypsinized, washed, and the percentage of PKH26$^+$ cells was determined using an Attune NxT Acoustic Cytometer (Thermo Fisher Scientific). DAPI (Sigma) was used to exclude dead cells with laser VL1.

**Senescence assay.** β-Galactosidase activity was measured in Gluc-CC y Gal-CC according to the manufacturer's instructions (Cat no. 94433, Sigma). β-Galactosidase$^+$ cells were determined using an Axiovert 135 TV optical microscope (ZEISS, Germany), and the ImageJ 2.0.0 software (NIH, USA) was used for further analysis.

**Chemoresistance in vitro assays.** PDAC cells were seeded in a 12-well plate at 10$^6$ cells/well. Following 14 days of culture with glucose or galactose, Gal-CC or Gluc-CC cells were treated with 1 μg/ml of GEM, 3 μM of MTX, or 1 μM of DXR for 2 days. After 2 days, drugs were removed and PDAC cells were allowed to recover for 3 days. For metabolic drugs, Gluc-CC and Gal-CC were treated with 5 μM of Menadione, 10 nM of Rotenone, 100 μM of Resveratrol, 3 mM of Metformin, or 5 mM of 2-DG for 3 days. Following treatments, the supernatant was

collected, and the ToxiLight™ Bioassay Kit (Cat no. LT07-217, Lonza, Switzerland) was used to evaluate the toxicity using a bioluminescence GLOMAX Luminometer (Promega).

**Measurement of mitochondrial metabolism by Seahorse analyzer.** Measurements of OCR were performed with a XF extracellular flux analyzer (Seahorse Bioscience, Agilent Technologies, Santa Clara, CA, USA), and data were collected with the XF96 1.4.2 Software (Agilent). Briefly, 14-day-old Gal-CC or Gluc-CC were trypsinized and re-seeded at a density of 10$^5$ cells per well in a XF24 cell culture microplate with DMEM (4.5 g/L glucose, +L-Glutamine, -Pyruvate, pH 7.6 at RT) supplemented with 10% heat-inactivated FBS (Gibco) and 1% Pen/Strep. On the day of the assay (24 h after re-seeding), medium was changed to non-carbonated Seahorse measurement medium and the XF24 plate was transferred to a temperature-controlled (37 °C) XF24 Extracellular Flux analyzer and equilibrated for 10 min. To determine the basal respiration, four assay cycles (1-min mix, 2-min wait, and 3-min measuring period) were performed. Then oligomycin (4 μM) was added by automatic pneumatic injection (three assay cycles) to inhibit ATP synthase and thus approximate the proportion of respiration used to drive ATP synthesis versus proton leak-linked respiration. Oligomycin was followed by an injection of FCCP (carbonyl cyanide p-trifluoromethoxyphenylhydrazone) (0.5 μM) to completely dissipate proton motive force and maximally stimulate mitochondrial respiration (three assay cycles), thus determining SRC and substrate oxidation capacity. An injection of rotenone (4 μM) and antimycin A (2 μM) was used to correct for the non-mitochondrial respiration rate (three assay cycles), which was subtracted from all the other rates. Basal respiration was calculated as the oligomycin-sensitive fraction of mitochondrial respiratory activity, estimating the proportion of basal respiration used to drive ATP synthesis. To determine ECARs, experiments were terminated by injection of 2-DG (100 mM), correcting for non-glycolytic acidification. Injection of 2-DG enables the calculation of glycolytic acidification. Data obtained after oligomycin treatment induced glycolytic capacity. Raw data were normalized to total cell numbers, determined by optical absorbance of lysed crystal violet-stained cells.

**RNA sequencing analysis.** Total RNA was isolated by the GTC method using standard protocols[49]. Sequencing libraries were prepared with the "NEBNext Ultra II Directional RNA Library Prep Kit for Illumina" (NEB no. E7760) as recommended by the kit manufacturer. Briefly, polyA+ fraction was purified and randomly fragmented, converted to double-stranded cDNA, and processed through subsequent enzymatic treatments of end-repair, dA-tailing, and ligation to adapters. Adapter-ligated library was completed by PCR with Illumina PE primers. The resulting purified cDNA libraries were applied to an Illumina flow cell for cluster generation and sequenced on an Illumina NextSeq 550 (with v2.5 reagent kits), according to the manufacturer's protocols. RNAseq data sets were analyzed using the tool Nextpresso[51]. Nextpresso is comprised of four basic levels: (1) quality check, (2) read cleaning and/or down-sampling, (3) alignment, and (4) analysis (gene/isoform expression quantification, differential expression, gene set enrichment analysis, and fusion prediction). The gene signatures (Hallmark genesets, h.all.v7.1.symbold.gmt) from GSEA—Molecular Signature Database for Gene set enrichment analysis was used for pathway enrichment analysis with GSEA v4.0.3 (Broad Institute).

**Zebrafish maintenance and xenograft assays.** Zebrafish embryos were obtained by mating adult zebrafish (*Danio rerio*, wild type), maintained in 30-l tanks with a ratio of 1 fish per liter of water, with 14 h/10 h light/dark cycle and a temperature of

28.5 °C according to published procedures[35]. All the procedures used in the experiments as well as fish care were performed in agreement with the Animal Care and Use Committee of the University of Santiago de Compostela and the standard protocols of Spain (Directive 2012-63-UE). At the final point of the experiments, zebrafish embryos were euthanized by tricaine overdose.

For zebrafish xenograft assays and image analysis, zebrafish embryos were collected at 0 h post-fertilization (hpf) and incubated until 48 hpf at 28.5 °C. At 48 hpf, hatched embryos were anesthetized with 0.003% of tricaine (Sigma). mCherry-H2B-labeled PANC185scd and PANCA6L Gluc-CC and Gal-CC were trypsinized, resuspended, and concentrated in an Eppendorf at $10^6$ cells per tube for each condition. Cells were then resuspended in 10 µl of PBS with 2% polyvinylpyrrolidone to avoid cellular aggregation. Borosilicate needles (1 mm O. D. × 0.75 mm I.D.; World Precision Instruments) were used to perform the xenograft assays in the zebrafish embryos. Between 100 and 150 cells were injected into the circulation of each fish (Duct of Cuvier) using a microinjector (IM-31 Electric Microinjector, Narishige) with an output pressure of 34 kPA and 30 ms of injection time per injection. Subsequently, the injected embryos were incubated at a temperature of 34 °C for 6 dpi in 30 ml Petri dishes for each condition with salt dechlorinate tap water. Imaging of the injected embryos was performed using a fluorescence stereomicroscope (AZ-100, Nikon) at 1, 4, and 6 dpi in order to measure the proliferation of the PANC185scd- and PANCA6L-injected human cancer cells inside the zebrafish circulation in each of the conditions assayed.

The image analysis of the injected embryos was carried out using the Quantifish software v2.1 (University College London, London, UK) in order to obtain the proliferation ratio of the cells in the region of the caudal hematopoietic tissue of the embryos, where the cells proliferate and metastasize. This program measures in each of the images provided the intensity of the fluorescence and the area of the positive pixel above a certain threshold of the cells. With these parameters, an integrated density value is obtained allowing the researcher to compare different times between images to reach a proliferation ratio.

**Lactate production assay**. Supernatant from Glu-CC and Gal-CC were collected to evaluate the changes in the levels of lactate production. The analysis was performed using the Lactate Assay Kit (Sigma Aldrich, St. Louis, MO, USA) according to the manufacturer's instructions. The optical density was determined using a Synergy™ HT Multi-Mode Microplate Reader (BioTek, Winooski, VT, USA) at a wavelength set to 570 nm.

**ATP determination assay**. Lysate pellets of cells from Glu-CC and Gal-CC were collected to evaluate the changes in the levels of ATP. The analysis was performed using the Molecular Probes ATP Determination Kit (Thermo Fisher Scientific, Waltham, MA, USA) according to the manufacturer's instructions. Bioluminescence was determined using a Synergy™ HT Multi-Mode Microplate Reader (BioTek, Winooski, VT, USA).

**Isolation of macrophages and T cells from human blood**. Blood samples from healthy donors were provided by the BioBank Hospital Ramón y Cajal-IRYCIS (PT13/0010/0002), integrated in the Spanish National Biobanks Network (ISCIII Biobank Register No. B.0000678). Samples were processed following standard operating procedures with the appropriate approval of the Hospital Ramón y Cajal Ethical and Scientific Committee (Control No.: DE-BIOB-73 AC65, RG.BIOB-57, and RG.BIOB-54), with informed consent and according to Declaration of Helsinki principles. Blood samples were diluted with PBS (Cat no. 10010023, Gibco) and Ficoll (Cat no. L6115, Merck) was used to isolate peripheral blood mononuclear cells (PBMCs). PBMCs were divided across three 6-well culture plates (per donor) with RPMI 1640 media containing 10% FBS and 50 units/ml penicillin/streptomycin. After 24 h, monocytes adhered to the plate surface were separated from the lymphocytes that remain in suspension and that were seeded in T-75 flasks for their subsequent activation and growth.

**Cytokine array**. Changes in protein levels related to immune evasion processes were determined by measuring changes in the levels of cytokines or inflammatory-related molecules released into the culture medium by Gluc-CC and Gal-CC cells, using the Proteome Profiler Human Cytokine Array Kit (R&D Systems, Minneapolis, MN, USA), according to the manufacturer's instructions. Cytokine array images were obtained using the MyECL Imager (Thermo Fisher Scientific). For densitometry analysis, array images were analyzed using PhotoShop CS4 EXTENDED V11.0 (Adobe Systems, USA). Data were normalized to the indicated array positive control.

**TFGβ secretion assay**. Culture medium from Gluc-CC and Gal-CC were collected to evaluate the changes in the levels of TGFβ secretion. The analysis was performed using the Human TGFβ1 Immunoassay Quantikine ELISA Kit (R&D Systems, Minneapolis, MN, USA) according to the manufacturer's instructions. The optical density was determined using a Synergy™ HT Multi-Mode Microplate Reader (BioTek, Winooski, VT, USA) at a wavelength set to 450 nm.

**Macrophage immunoevasion assay**. Five days after isolation of macrophages from the blood of healthy donors (see above), macrophages were labeled with the red membrane dye DilC, according to the manufacturer's instructions (Cat no. D3911, Thermo Fisher). Gluc-CC and Gal-CC were harvested and labeled with PKH67 green fluorescent cell membrane labeling dye (Sigma) according to the manufacturer's instructions. Labeled cells were seeded together with labeled macrophages during 24 h. Afterwards, macrophage-mediated phagocytosis (DilC +/PKH67+ cell percentages) was determined using the AttuneNxT cytometer (Thermo Fisher Scientific), and data were analyzed with the FlowJo 9.3 software (TreeStar, Ashland, OR).

**T cell immunoevasion assay**. Seven days after isolation and expansion of T cells from the blood of healthy donors, T cells were activated with phytohemagglutinin-P (Cat no. L8754, Merck) at a concentration of 5 µg/ml and with 20 ng/ml of human rIL-2 (Cat no. PHC0026, Gibco) for 7 days. Gluc-CC and Gal-CC were harvested and co-seeded with activated T cells during 10 days. Afterwards, T cells were removed and the extent of cytotoxicity was determined with the ToxiLight™ Bioassay Kit (Cat no. LT07-217, Lonza, Switzerland) using a bioluminescence GLOMAX Luminometer (Promega). The data were analyzed with the Prism 8 software (San Diego, CA).

**Statistical analysis and reproducibility**. Results are presented as means ± standard deviation (stdev) or ±standard error of the mean (SEM), as indicated in the figure legends. Different two-sided statistical analyses were performed: one-way analysis of variance (ANOVA; two sided) with Bonferroni adjustment or with an unpaired two-tailed $t$ test to determine significance, two-way ANOVA with Tukey's multiple comparisons test, or unpaired two-sided (confidence interval of 95%) $t$ test for multiple comparisons using the Holm–Sidak method were used to determine differences between means of groups. $p$ Values < 0.05 were considered statistically significant. GraphPad software (GraphPad Prism 8.0, GraphPad Software, La Jolla, CA, USA) was used to perform the statistical analysis of the data.

The number of biologically independent samples are indicated in the figure legends. Repeated independent experiments per each panel with similar results are shown below. n = 1 (Figs. 1a, e, 2b, 3d, l, 4a, k, 5c, k, 6e, n, 7g, k and Supplementary Figs. 1a, c, h, i, 2a–e, h–j, m, 4a, b, 3e, 5a, d–f, 6a, b, 7a–d); n = 2 (Figs. 1f, g, 2e–j, l, m, 3e, 5a, 6c, d, 7f, j and Supplementary Figs. 2a–e, g, 3b, d, 5b, h–k); n = 3 (Figs. 2f, 6b, 7b, d and Supplementary Figs. 1f, 3i, 6d–f), n = 4 (Figs. 2d, 3b and Supplementary Figs. 1b, 2f).

**Reporting summary**. Further information on research design is available in the Nature Research Reporting Summary linked to this article.

## Data availability
RNAseq data from Gluc-CC and Gal-CC cells, generated in this study, have been deposited in the ArrayExpress database[52] at EMBL-EBI (www.ebi.ac.uk/arrayexpress) under accession number E-MTAB-9483. Source data are provided with this paper.

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

## Acknowledgements

We acknowledge and thank Dr. Nuria Malats and Jaime Villarreal from the Spanish National Cancer Research Center (CNIO) for RNA sequencing and analysis, funded by Fondo de Investigaciones Sanitarias (FIS) grant PI18/01347. We thank Patricia Sánchez-Tomero and Marina Ochando-Garmendia for technical assistance and support and Dr. Raúl Sánchez Lanzas for assistance with autophagy experiments. We want to particularly acknowledge the patients and the BioBank Hospital Ramón y Cajal-IRYCIS (PT13/0010/0002) integrated in the Spanish National Biobanks Network for its collaboration and, in particular, Adrián Povo Retana for macrophage isolation. We would also like to thank the Transmission Electron Microscopy Unit Laboratory, part of the UAM Interdepartmental Investigation Service (SIdI); Coral Pedrero for exceptional help with in vivo experiments; and the laboratories of Dr. Amparo Cano and Dr. José González Castaño for reagents and helpful discussions. S.V. was a recipient of an Ayuda de Movilidad del Personal Investigador del IRYCIS, a mobility grant from the Instituto Ramón y Cajal de Investigación Sanitaria (IRYCIS), Madrid, Spain, and a pre-doctoral fellowship from the Comunidad de Madrid, Ayudas Para La Contratación De Investigadores Predoctorales Y Posdoctorales (PEJD-2017-PRE/BMD-5062), Madrid, Spain. This study was supported by a Rámon y Cajal Merit Award (RYC-2012-12104) from the Ministerio de Economía y Competitividad, Spain (to B.S.); funding from la Beca Carmen Delgado/Miguel Pérez-Mateo from AESPANC-ACANPAN Spain (to B.S.); a Conquer Cancer Now Grant from the Concern Foundation (Los Angeles, CA, USA) (to B.S.); a Coordinated grant (GC16173694BARB) from the Fundación Asociación Española Contra el Cáncer (AECC) (to B.S.); FIS grants PI18/00757 (to B.S.), PI16/00789 (to M.A.F.-M.), PI18/00267 (to L. G.-B.); co-financed through Fondo Europeo de Desarrollo Regional (FEDER) "Una manera de hacer Europa"); a Miguel Servet award (CP16/00121) (to P.S.); a Max Eder Fellowship of the German Cancer Aid (111746) (to P.C.H.); and the German Research Foundation (DFG, CRC 1279 "Exploiting the human peptidome for Novel Antimicrobial and Anticancer Agents"; to P.C.H.).

## Author contributions

S.V. acquired, analyzed, and interpreted all data as well as developed the study concept and drafted the manuscript; S.A., L.M.-H., D.N., and M.A.-N. performed in vitro experiments and analyzed data; L.Y. and L.R.-C. performed in vivo studies; J.A.R. performed TEM experiments; P.C.-S. and L.S. designed and performed the zebrafish studies; E.R.M. and L.G.-B. performed the Seahorse studies and IHCs; K.W., K.T., and P.C.H. performed, designed, and analyzed the Nanog-YNL studies; E.G.-A. is our in-house pathologist and assisted with in vivo histological analyses; P.C.H. and C.H. produced the 185scd cell line. P.S. provided expertise on CSC metabolism and assisted in the execution and analysis of key in vitro experiments; M.A.F.-M. and B.S. developed the study concept, obtained funding, interpreted the data, and drafted/edited the manuscript; and all authors edited the manuscript.

## Competing interests

The authors declare no competing interests.
