## [Peer Review File · Nature Communications]

Point-by-point response to NCOMMS-19-99997-T

Reviewer #1 (Remarks to the Author):

In sum, the experimental data are clear and the application of galactose to enrich for PaSCs is a clever strategy with potential to be a useful tool in pancreatic cancer and other cancers driven by stem cells. Accordingly, this work provides a useful “method”, as opposed to a study, as it does not address the fundamental biological question of why PaSCs increase their mitochondrial content. That said, it would still benefit from clarification of two major concerns.

Authors comment: We thank Reviewer no1 for his/her positive assessment of the work, specifically the importance of the system as a tool for studying pancreatic CSCs as well as CSCs of other tumor entities.

Major concerns:

1) First, the "Discussion" section starts out by stating that culture in galactose enriches PaSCs and that non-PaSCs cannot survive the absence of glucose. Data are not provided that support either of these claims. More importantly, this is a central question about the method. Does galactose treatment enrich for PaSCs by killing off, or attrition (lack of proliferation), of non-PaSCs? Alternatively, non-PaSCs may increase their mitochondrial mass, etc. and become PaSCs. Data to clarify this point are critical to understand how the culture conditions work and the nature of PaSCs.

Authors comment: We agree with the reviewer regarding this point. In fact, this same point was brought up by Reviewer no.2. Indeed, answering this question was difficult but necessary to try and determine whether the enrichment in PaCSCs is a consequence of 1) non-PaCSCs being killed off, 2) non-PaCSCs converting in PaCSCs or 3) both. To address this point we have, on the one hand, highlighted better the cell death that is observed in galactose-treated cultures, resulting in a culture that is 30-40% the confluency achieved in glucose-treated cultures. Thus, cell death occurs, reducing proliferation and coincides with an enrichment in the PaCSC population. We refer the review to the re-written section “**Galactose promotes sustained enrichment of PaCSCs**” on **Page 4-5**. On the other hand, we sorted Autofluorescent-negative and CD133-negative populations away from the bulk population of cells and placed the negative fractions in galactose or in glucose. Regardless of the carbon source, CSC-biomarker-negative cells gave rise to CSC-biomarker-positive cells (**Supplementary Figure 2a-b**). However, at times > 4 d post-sorting, a sharp decrease in the percentage of CSC-biomarker-positive cells was observed in Gluc-CC, while in Gal-CC the percentages either remained constant or increased, and at d 11 were ~2-fold higher than Gluc-CC (**Supplementary Figure 2a-b**). This fold-difference coincided with an ~2-fold decrease in cell viability in Gal-CC, confirming that PaCSC enrichment coincides with non-PaCSC cell death (**Supplementary Figure 2c**). Importantly, when CD133+ sorted cells were directly cultured in galactose, cell viability was not affected (**Supplementary Figure 2d**). Taken together, we concluded that in galactose-containing media 1) PaCSCs survive, 2) a large percentage of non-PaCSCs die, but 3) a smaller percentage of non-PaCSCs convert into PaCSCs (i.e. plasticity). The sum of these biological consequences results in a PaCSC-enriched culture. We have not been able to determine the mechanism(s) behind this plasticity, but discuss possibilities, such as those mentioned by the reviewer (e.g. non-PaSCs may increase their mitochondrial

mass) in the re-written Discussion section. Deciphering how exactly non-PaCSCs convert into PaCSCs in galactose culture conditions is part of another incomplete study underway in the laboratory.

2) Second, while the phenotypic data presented for galactose-cultured cells are all consistent with a PaSC state, they are modest, often being less than 2-fold. This is certainly not a problem, but it begs the question of how the data compare to more classical PaSC enrichment strategies, e.g. CD133+ sorting. Having data using classic enrichment strategies as a positive control, in at least the functional assays (e.g. tumor growth), would provide important context as a basis for comparison to evaluate the utility of this method.

Authors comment: We understand the reviewer's comment, and thus have provide comparison data in the form of a table where the CSC frequencies have been calculated for autofluorescent+ cells and CD133+ cells from in vivo ELDA assays (**Figure 2m versus Supplementary Figure 2m**). Comparing Gal-CC to more traditional methods, the Gal-CC approach is similar or perhaps more efficient in enriching for CSCs.

Furthermore, we provide additional comparative analyses. First, we provide data showing CSC marker enrichment levels (fold-increase) in spheres versus adherent cultures (**Supplementary Figure 1g**). Second, we show the number of sphere/ml formed comparing CD133+ to CD133- sorted cells (**Supplementary Figure 2e**). Third, we show the fold-increase in pluripotency genes expression (*KLF4*, *OCT3/4*, *SOX2* and *NANOG*) for sph compared to adherent cultures (**Supplementary Figure 2f**). Lastly, we reference our published analysis of lncRNAs in sph compared to adherent cultures. As can be seen in all of the aforementioned figures panels, CSC enrichment in Gal-CC is equal to or better than more traditional approaches, such as CD133-sorting or sphere formation. We thank the reviewer for requesting this additional and important data, and we feel that its addition considerably improves the study and supports the use of galactose as a means to enrich for PaCSCs compared to other more traditional methods that are more laborious or technically more difficult.

Lastly, we would like to point out the new **Figure 4A**, which shows gene set enrichment analysis in Gal-CC versus Gal-CC samples. As can be seen, Gal-CC upregulate many pathways involved in stemness.

Minor comments/suggestions

1. Glu is the abbreviation for the amino acid glutamate. I recommend using Glc for glucose. **Authors comment:** We have changed Glu for Gluc in the manuscript

2. Foundational work by Viale and Draetta on PaSCs was not and should be cited.

Authors comment: We thank the reviewer for pointing out important work from Viale and Draetta on PaSCs. Several articles from these authors have been included.

3. In Fig.1H, anaerobic glycolysis is not an accurate description for the pathway in the blue box. **Authors comment:** This has been corrected.

4. In Fig.2M, there is a typo in the table (10,000 = 1,000).

Authors comment: This has been corrected.

5. This reviewer prefer bar graphs (or an alternative more concise format) in place of flow histograms.

Authors comment: For all of the representative flow histograms, we have provided accompanying bar graphs. Thus, both are provided. We feel that it is important to show the flow histograms so that the reviewer and readers can evaluate our gating

strategy and criteria for “positive populations”. These histograms are merely representative and the data upon which the statistical significance was calculated are shown in the accompanying bar graphs.

Reviewer #2 (Remarks to the Author):

Based on the comments raised by Reviewer no.2, his/her main concern is that we have not sufficiently proven that the 2D galactose-bases system enriches for PaCSCs. We have better explained our results and included additional key experiments that we believe will now convince the reviewer that our system indeed enriches for CSCs. We thank the reviewer for allowing us to address his/her concerns.

1) The data are a bit surprising as galactose has long been used to reduce numbers of neural stem cells in vivo resulting in senescence. Admittedly these are benign stem cells rather than tumour initiating cells so this inconsistency could be explainable.

Authors comment: The reviewer is correct. It has been shown that stem cells use glycolysis and not OXPHOS to meet their energy requirements. Thus, placing neural stem cells in a glycolysis-limiting environment, that is galactose conditions, would promote their reduction or death. However, PaCSCs use OXPHOS to meet their energy requirements, something we have already published (Sancho et al, Cell Metab. 2015 Oct 6;22(4):590-605). Thus, in galactose conditions an enrichment is observed because cells in galactose are “obligated” to use OXPHOS. We have emphasized this point in the revision to avoid confusion.

The difference is therefore cell type specific (normal stem cell vs CSC), but at the same time an important point of the paper and one that has allowed us to utilize CSC metabolism as a means to enrich for PaCSCs.

2) More worrisome is whether or not the data as presented actually support the authors’ conclusions that they are actually enriching PaCSCs, particularly with the data presented in the first and second figures, which call into question the data presented in the remaining figures. For this reason, I do not believe the paper is publishable in its present form. Should the authors be able to respond to my queries about the first two figures I would be happy to review the data in the remaining figures as these are dependent on proper enrichment of PaCSCs.

Authors comment: Again, this reviewer’s concern is in line with Reviewer #1’s Major Point 1. As described above, answering this question was difficult but necessary to try and determine whether the enrichment in PaCSCs is a consequence of 1) non-PaCSCs being killed off, 2) non-PaCSCs converting in PaCSCs or 3) both. To address this point we have, on the one hand, highlighted better the cell death that is observed in galactose-treated cultures, which results in a culture that is 30-40% the confluency achieved in glucose-treated cultures. Thus, cell death occurs, reducing proliferation and coincides with an enrichment in the PaCSC population. We refer the review to the re-written section “**Galactose promotes sustained enrichment of PaCSCs**” on **Page 4-5**. On the other hand, we sorted the Autofluorescent-negative and CD133-negative population away from the bulk population of cells and placed the negative fractions in galactose and in glucose. Regardless of the carbon source, CSC-biomarker-negative cells gave rise to CSC-biomarker-positive cells (**Supplementary Figure 2a-b**). However, at times > 4 d post-sorting, a sharp decrease in the percentage of CSC-biomarker-positive cells was observed in Gluc-CC, while in Gal-CC the percentages either

remained constant or increased, and at d 11 were ~2-fold higher than Gluc-CC (**Supplementary Figure 2a-b**). This fold-difference coincided with an ~2-fold decrease in cell viability in Gal-CC, confirming that PaCSC enrichment coincides with non-PaCSC cell death (**Supplementary Figure 2c**). Importantly, when CD133+ sorted cells were directly cultured in galactose, cell viability was not affected (**Supplementary Figure 2d**). Taken together, we concluded that in galactose-containing media 1) PaCSCs survive, 2) a large percentage of non-PaCSCs die, but 3) a smaller percentage of non-PaCSCs convert into PaCSCs (i.e. plasticity). The sum of these biological consequences results in a PaCSC-enriched culture. We have not been able to determine the mechanism(s) behind this plasticity, but discuss possibilities, such as those mentioned by the reviewer (e.g. non-PaCSCs may increase their mitochondrial mass) in the rewritten Discussion section. Deciphering how non-PaCSCs convert into PaCSCs in galactose culture conditions is beyond the scope of this study.

In addition, as to further convince the reviewer that we have indeed enriched for CSCs, we have provided additional data comparing our results with those obtained using published and traditional CSC enrichment methods (e.g. CD133 sorted cells or spheres cultures). First, we provide data showing CSC marker enrichment levels (fold-increase) in spheres versus adherent cultures (**Figure 2d versus Supplementary Figure 1g**). Second, we show the number of sphere/ml formed comparing CD133+ to CD133-sorted cells (**Figure 2e versus Supplementary Figure 2e**). Third, we show the fold-increase in pluripotency genes expression (*KLF4*, *OCT3/4*, *SOX2* and *NANOG*) for sph compared to adherent cultures (**Figure 2i versus Supplementary Figure 2f**). Fourth, we reference our published analysis of lncRNAs in sph compared to adherent cultures. Lastly, we have provided a table where the CSC frequencies have been calculated for Galactose cells and for CD133+ and Autofluorescent+ cell from in vivo ELDA assays (**Figure 2m versus Supplementary Figure 2m**). As can be seen in all of the aforementioned figures panels, CSC enrichment in galactose-cultured cells is equal to or better than more traditional approaches, such as CD133-sorting or sphere formation, indicating that we have indeed enriched for PaCSCs at the phenotypic and functional levels.

Lastly, we would like to point out the new **Figure 4A**, which shows gene set enrichment analysis in Gal-CC versus Gal-CC samples. As can be seen, Gal-CC upregulate many pathways involved in stemness, again supporting that in galactose, we are indeed enriching for CSCs.

General comment

After several read throughs the identity of the PDAC PDXs that were used is still not clear to me. The methods state that these cells were obtained from Dr. Manuel Hidalgo under an MTA with the CNIO, but are not more specific. Even a reference to a prior paper describing these PDXs would be helpful as this would validate, at a minimum, that these cells came from a pancreatic source. Please correct me if this is specified elsewhere in the manuscript.

Authors comment: The reviewer is absolutely correct and we regret this oversight. A reference and additional information have been provided in the Materials and Methods.

Figure comments

Figure 1.

a-e. These data are solid and it is quite clear that Gal-CCs have higher amounts of mitochondria with substantially different morphology.

Authors comment: This is an underlying observation of the paper that we have validated from different angles and using different approaches.

g. I haven't seen DAPI used as a marker of nuclear DNA dilution. Usually this is done with dyes such as CFSE. If the authors intend to use this they should provide a reference validating it.

Authors comment: We agree with the Reviewer that this data was confusing and the use of DAPI as a marker of nuclear DNA dilution may be questionable. Thus, since this observation is not of extreme relevance, we have removed it from the paper so as not to confuse the readers.

Figure 2.

a-b. The enrichment of the markers does not appear to be proportionate. Autofluorescence goes up ~5 fold, but CD133 is enriched 3-fold, and CXCR4 only 2-fold. Would not all of these markers be expected to rise in concert with PaCSC enrichment? These do not appear to be consistent with panel b in which the enrichment of all markers is ~3 fold.

Authors comment: The author is correct that the data presented in Figure 2d did not correlate with the representative plots due to computing errors. We apologize for this oversight and thank the Reviewer for catching this error.

Regarding marker expression in general, we feel it is important to note that CSC marker enrichment is not black and white. We have modified the Discussion to discuss this point and now state "Not surprisingly, we did not obtain 100% CSC biomarker staining in Gal-CC, which we reconcile by acknowledging that heterogeneity exists within the CSC population(s), making it impossible to detect all CSC clones as not all CSCs (or OXPHOS-dependent hybrid CSCs) will express the same single biomarker, or combinations of different biomarkers. Nonetheless, the increase observed in defined CSC markers (>15%) and others (mitochondrial markers of approx. 100% and immune evasion markers of >50%) clearly indicate an enrichment over that observed with traditional approaches^{6,7,13}."

To date no single technique provides long term 100% stable enrichment of CSCs, but when compared to Figure 1A, cells cultured in galactose are stably enriched in PaCSCs. While not all markers reach 100%, the system as a whole highlights a tremendous advancement.

c. The cells state in galactose, as described in the text. The authors do not state that they observe a die-off of the remaining cells and the data support this. This leaves the possibility that these cells are converted to CD133+ PaCSCs but does not support an enrichment of PaCSCs.

Authors comment: This very important point has been addressed above in our response to the Reviewer's 2nd Main Point.

h. Why are double-positive CD133+ and CXCR4+ cells used here when CD133+ was deemed sufficient previously?

Authors comment: We often combine markers, and it has been shown that CD133/CXCR4 double staining better detects the CSC population in PDAC cultures. Thus, this is why we used this double staining approach here. Moreover, the number

of individual cells achieved from these sphere cultures were low in number and the cytometry results obtained with the double staining was more reliable.

k. While tumour initiation was increased in mice bearing Gal-CCs it is not clear from the methods or from the text if this is statistically significant. The authors state that 5-10 mice per group were used and that no statistical methods were used to predetermine sample size. Assuming that $n=5, 6, 7, 8, 9,$ or 10 per group with groups of equal sizes, even the most significant limiting dilution of Glu-CC vs. Gal-CC (1000 cells/injection) at which the greatest difference in tumour initiation is observed ($0/n$ in Glu-CC vs $4/n$ in Gal-CC) does not reach statistical significance by Fisher's Exact Test. This makes it more difficult to accept these results which are critical to the authors' claim that they have enriched PaCSCs.

Authors comment: For the in vivo tumorigenicity studies, 8 injections were performed. The data was then uploaded to the ELDA: Extreme Limiting Dilution Analysis platform (<http://bioinf.wehi.edu.au/software/elda/>) and the statistical analysis was performed. This platform is routinely used to calculate CSC frequency and significance, and we respectfully disagree with the reviewer that the data are not significant. The analysis takes into consideration tumor take across all dilutions, not just one.

Likewise, the number of injections performed are in fact greater than what is typically published. Moreover, publications usually include 3 dilutions, while here we have included 4 dilutions (50,000, 10,000, 5,000 and 1,000). **Nonetheless**, we have performed a new ELDA repeat experiment to increase the " n " to $n=12$ for each dilution. This has improved the overall statistical significance of the entire in vivo study. We have provided the analysis generated using the ELDA software. We hope these additional data now satisfy the reviewer's concerns.

The value of the confidence choice entered was "0.95"

The value of the observed choice is "TRUE"

The value of the test_unit_slope choice is "TRUE"

The value of the test_difference choice is "TRUE"

Limiting dilution data entered.

Counter	Dose	Tested	Response	Group
1	50000	12	5	A
2	10000	12	5	A
3	5000	12	1	A
4	1000	12	1	A
5	50000	12	10	B
6	10000	12	10	B
7	5000	12	6	B
8	1000	12	5	B

The number of lines of data entered = 8

Confidence intervals for
1/(stem cell frequency)

Group	Lower	Estimate	Upper
A	94760	51521	28012
B	16776	10075	6051

Overall test for
differences in stem
cell frequencies
between any of the
groups

Chisq	DF	P.value
22.5	1	2.12e-06

Reviewer #3 (Remarks to the Author):

General comments

1) Despite manuscript is about a novel 2D culture enriched with PaCSC, authors do not put too much emphasis in the establishment of the right culture conditions:

– Why 0.9 gr/L of Glu or Gal and no other concentrations?

Authors comment: In 1925, Otto Warburg (Warburg, O. (1925) The metabolism of carcinoma cells. *J. Cancer Res.* 9, 148–163) was able to demonstrate that osteosarcoma cells utilize 18-fold less glycolysis when galactose is used as a sugar source instead of glucose. It is now understood that galactose leads to activation of mitochondrial OXPHOS due to the energetic demands of galactose breakdown through glycolysis. Utilization of galactose through glycolysis results in a net production of zero ATP, leading to a compensatory up-regulation of OXPHOS in the presence of exogenous mitochondrial OXPHOS substrates, such as pyruvate and glutamine. Thus, 0.9g/L (5mM) Glucose replacement with 0.9g/L (5mM) Galactose in the presence of mitochondrial substrates pyruvate and glutamine is a **known and commonly used strategy to induce mitochondrial OXPHOS** (e.g. *J Biol Chem.* 2018 Oct 12;293(41):16019-16027). Likewise, sugar concentrations of 5mM are used in biological systems, to recapitulate physiological levels of glucose (5mM) and to avoid any artifacts caused by supraphysiological levels of glucose.

Thus, we did not place emphasis on why we chose these concentrations as they are well established concentrations in the literature. Nonetheless, we have added more details and references to the Results section and the Materials and Methods section to highlight these points. We hope that this satisfies the reviewer's concern regarding the sugar concentrations used.

– Why Pa-SCS were cultured for 7- 14 days (as described in methods sections and some figure legends) for then being used for analysis and not more or less days? Did authors check in a time manner an increase of the PaSCS to select these timelines?

Authors comment: As described in the text, 14 days was chosen based on maximum CSC marker expression across all cultures evaluated. This has been stressed in the text. More days (30 days) were also analyzed and the same results were obtained. Regarding the latter, we refer the reviewer point 6 below, where we established Gal-CC from a digested PDX, which was cultured for 63 days.

2) Can PaSCS that have been cultured with galactose be cryopreserved and thawed again to perform new assays?

Authors comment: Yes, Gal-CC cultures can be trypsinized, and re-plated in glucose or cryopreserved for subsequent thawing and reuse. We have stated this in the Materials and Methods section.

3) Authors are not presenting any genomic analysis of the cell cultures. How heterogeneous, in terms of somatic mutations for example, is the population of PaCSC for each model?

Authors comment: It is a fact that during in vitro passaging any cell within the culture (including PaCSCs) can acquire additional somatic mutations, a phenomenon that occurs in all in vitro culture. However, performing whole genome sequencing on PaCSCs versus non-PaCSCs isolated from the different PDX models utilized is beyond the scope of this study, and would likely not provide relevant information as we have shown that in cultures with dominant PaCSC clones, these CSCs are essentially genetically identically to the non-CSC population. Relevant differences can be found at

the epigenetic level, as we have previously shown (Zagorac et al, Cancer Res. 2016 Aug 1;76(15):4546-58). Likewise, we believe that since we were able to establish PaCSC-enriched Gal-CC for 4 different PDX-derived cells cultures with varying heterogeneity, difference in the somatic mutation profile of the PaCSC population (within a single pDX and across PDXs) are likely not playing a relevant role.

Nonetheless, it is important to note that these PDX tumors have been previously characterized at the genetic level (Martinez-Garcia, et al. Genome Med. 2014 Apr 16;6(4):27), and recently we confirmed that all tumors harbor *KRAS* mutations and differing mutations in *Tp53* (D'Errico et al, Oncogene. 2019 Jul;38(27):5469-5485). Across the different PDX-derived cultures there is heterogeneity, but the common PDAC driver mutations are present.

In addition, believe it would be more interesting and valid to perform RNAseq analysis on the Gluc-CC versus Gal-CC cultures to better understand the transcriptional landscape of these cells and have thus invested the resources to do so. We would like to point out the new **Figure 4A**, which shows gene set enrichment analysis in Gal-CC versus Gal-CC RNAseq samples. As can be seen, Gal-CC upregulate many pathways involved in stemness, again supporting that in galactose, we are indeed enriching for CSCs.

4) What is the view of the authors, do PDAC xenotransplant need to be isolated every time that Pa-SCS need to be generated, or can the 2D cell culture be generated and expanded in normal culture conditions (high glucose concentration) and then switch the culture medium to the galactose based to generate enough material for studying stemcellness?

Authors comment: Primary PDX-derived cultures were established as per standard protocols that we and others have published, including but not limited to: Nat Methods. 2014 Nov;11(11):1161-9, Cancer Res. 2014 Dec 15;74(24):7309-20, Gut. 2015 Dec;64(12):1921-35, Cell Metab. 2015 Oct 6;22(4):590-605, Cancer Res. 2016 Aug 1;76(15):4546-58.

No, PDXs do not need to be digested each time primary cultures with CSCs are required. Primary 2D cultures can be established and maintained for up to 15 passages without affecting the CSC compartment or percentages.

As detailed in the Methods section, "Low passage (< 15) PDX-derived primary PDAC cultures were trypsinized and seeded at a concentration of 800.000 cells in p100 plates with RPMI medium supplemented with 10% fetal bovine serum (FBS) and 50 units/mL of penicillin and streptomycin. After 24 hours, cells were cultured with either 1) glucose-free DMEM medium (Dulbecco's Modified Eagle Medium) (Thermo Fisher Scientific), supplemented with 5mM glucose (0.9 g/L), 10% FBS, 50 units/mL of penicillin and streptomycin and 1mM of pyruvate [Glucose: OXPHOS-independent conditions] or 2) glucose-free DMEM medium (Thermo Fisher Scientific) supplemented with 5mM galactose (0.9 g/L), 10% FBS, 50 units/mL of penicillin and streptomycin and 1mM of pyruvate [Galactose: OXPHOS-competent enriched conditions]."

5) Then, the main body of the manuscript is the characterization of these PaCSC, that have all the specific hallmarks known for pancreatic cancer stem cells. Therefore, this reviewer misses the further development of the possible applications of this 2D culture system for SCS-therapies, such:

– How efficient is this method to generate PaCSC following this method (did authors succeed on establishing 4 out of 4 or there were some failures?)

Of note, authors indicate in the manuscript the generation of 4 models but based on the naming of the samples, it seems two Gal-CC are generated from the same PDAC (PAN185 and PAN185scd), so the method has been tested in 3 different PDAC xenotransplant models. How reproducible, efficient is the establishment of this culture method?

– The main characterizations are shown for two of the four Gal-CC cultures. Therefore, more characterization is needed to evaluate the robustness of this protocol.

– In cancer, the expectations are that different tumors behave differently due to the complex and heterogenous landscape of tumor generation from different patients. Is it expected that PaCSC, from different patients, have the same behavior and for example all of them have the same response to the chemotherapies tested in this manuscript? The authors have not analysed enough number of samples to evaluate how relevant are the PaCSC generated under the same culture conditions regarding the patient response.

Authors comment: The PDX-derived cultures have been previously described and established. Thus, we did not succeed or fail, but merely chose 4 cultures for this study. The reviewer is correct that while 4 PDX-derived cultures were used throughout the study, many sub experiments were performed with only two in order to reduce the number of samples and related costs. Nonetheless, we do not feel that the use of 2 PDX-derived cultures for specific experiments is a limitation.

The 4 primary PDX-derived PDAC lines used have varying degrees of heterogeneity. All 4 have been previously established and described (Gastroenterology. 2009 Sep;137(3):1102-13, Cell Stem Cell. 2011 Nov 4;9(5):433-46 and Cancer Res. 2016 Aug 1;76(15):4546-58). Two heterogenous primary pancreatic tumors (PANC286 and PAN185), a liver metastasis (PANCA6L), and the PANC185 single cell-derived (scd) tumor that was generated by injecting a single autofluorescent-positive CSC isolated from PANC185 into an immunocompromised mouse (Cancer Res. 2016 Aug 1;76(15):4546-58). As we have previously shown, the heterogeneity should be highest in the primary tumors, less in the metastatic tumor and homogenous in the scd tumor (Cancer Res. 2016 Aug 1;76(15):4546-58). In fact, the 185scd culture allows for the study of multilineage differentiation, plasticity and CSC fate at the level of a single CSC clone, and allows one to ask the question of whether specific CSC features, such as metabolic plasticity, are a consequence of tumor heterogeneity where multiple different CSCs concomitantly contribute to the specific phenotype or an intrinsic property of CSCs independent of the degree of CSC heterogeneity (Cell Death Dis. 2013 Oct 17;4:e857). Since similar effects were seen between the 185scd clone and the other three cell lines, we can conclude that single-cell-cloned PaCSCs have the same plastic capacity and multilineage differentiation potential as cultures with more or less heterogenous CSC subpopulations. Thus, in theory, our 2D OXPPOS-based system could be applied to any primary PDAC culture regardless of the degree of CSC heterogeneity. The latter is important when contemplating the applicability or translatability of this culture system to personalized medicine or screening platforms that use cultures derived from PDXs or directly from CSCs isolated from freshly resected tumors. In both cases, the methodologies employed do not capture the complete heterogeneity of the original tumor, and this has created a debate as to whether ex vivo systems are biologically or clinically relevant. We can conclude from our studies that the metabolic plasticity of PaCSCs and the role of mitochondria in

PaCSC stemness is recapitulated across tumors of differing heterogeneity. It appears that mitochondria are one key feature that is homogenous and conserved across all CSC clones.

If requested by the reviewer **and approved by the editor**, we would be more than happy to include this extensive explanation in the Discussion section, **although this would significantly increase the text beyond 5000 words.**

Finally, and most importantly, while we feel that the additional details requested are very appropriate for a methods-based journal such as [REDACTED], the manuscript is no longer under consideration in [REDACTED], and we therefore believe that such minute details are no longer necessary. The goal of our study has always been to demonstrate that PaCSC-ness is linked to and depends on OXPHOS, PaCSC and be enriched for when this dependence is exploited, and the system established is amenable to many different biological assays, such as exploration of the role of immune evasion in PaCSC biology. Thus, we feel that the manuscript already touches upon and describes many pertinent biological aspects that the readership of *Nature Communications* will appreciate.

6) Is this culture method suitable for fresh tumor samples directly obtained from patients?

Authors comment: This is a very interesting point, however, as detailed in the introduction, few PDAC tumors are resectable. Those that are resectable are 1) small in size and 2) and highly fibrotic, often time containing less than 10% epithelial tumor cells. Many researchers have tried for years to establish primary cultures directly from surgically-resected PDAC tumors, but these attempts have failed and the current method for establishing primary PDAC cultures is to first establish a xenograft, from cultures can be subsequently established.

Therefore, as a proof of concept, we established galactose-PaCSC-enriched cultures directly from a freshly digested Panc185 PDX, shown in **Supplementary Figure 1h-i**. It is important to note that the kinetics of establishing a passage 0 culture from a digested tumor directly in galactose and glucose differs from a Gal- or Glu-CC derived from a “clean”, established and already passaged PDX-derived culture. The reviewer can note that in the first light micrographs from day 19 post digestion, small clones can be seen and murine fibroblasts are still present. Thus day 19 post digestion in galactose cannot be compared with day 19 Gal-CC derived from an established and already passaged PDX-derived culture. At day 33, however, cultures were fibroblast-free (i.e. “clean”) and CSC marker expression (i.e. autofluorescence) reached its maximum and did not significantly vary over the next 30 days.

7) Is this culture method scalable for screening for a larger number of compounds? In the manuscript it is evaluated the effect of 4 drugs, but what about 100 or 1000 compounds?

Authors comment: Yes, in theory the system can be scaled to screen hundreds to thousands of compounds; however, such a screening endeavor was not the goal of the study. We appreciate the reviewer’s interest in pointing out all of the possibilities for which the system could be used, but screening 100 or 1000 compounds would be better suited for another study, as the hits from such a screen would certainly open up new avenues of research, and better suited for follow-up publications.

Major points

1) In the abstract it is indicated that this manuscript describes a novel 2D in vitro method for long term enrichment of pancreatic CSC. This concept of long-term enrichment can be confusing, since the expectations of this reviewer would be that the culture of the PaSCS could be extended also in time, however the PaSCS are in culture for a maximum of 14 days (as described in methods section). Does it imply that PaCSC on galactose medium cannot be cultured longer, or authors have not tried? Considering that 3D culture system of adult stem cell allows long term expansion of pancreatic organoids (that include CSC and derived cells), what is the benefit of this 2D system?

Authors comment: We thank the reviewer for the comment. As stated in the beginning of the results section, “While 3D anchorage-independent spheres allow for CSC enrichment, they are not highly adaptable to various screening methodologies due to their non-adherent 3D-nature and the need to passage them once they reach critical mass. Attempts to establish 2D cultures enriched in CSCs using FACS have also proven ineffective as marker-enriched CSCs quickly re-establish the heterogeneity of the culture present prior to sorting.” Then in Figure 1A we show that FACS-sorted CSCs quickly differentiate into non-CSCs (e.g. CD133-negative) within 48 hours post sorting. Taken together, these observations and data show that maintaining a CSC-enriched culture is difficult to impossible in 2D. The same is true for organoids, that require specialized media and continuous passaging and to date there are insufficient studies to show the extent to which human PaCSCs are enriched and maintained in organoids. The Gal-CC methods is a 2D adherent system that does not require specialized media and cultures can be maintained without passaging for 14 days **or more**. While for our study there was no reason to culture cells beyond 14 days (the minimum time necessary to achieve maximum “stemness” from established PDX-derived cultures), if necessary or for a specific question or application, the cultures can be maintained for many months. Again, we refer the reviewer to **Supplementary Figure 1h-I**, where PDX-derived Gal-CC were maintained for 63 days (~2 months). Thus, based on the aforementioned reasons and the additional points highlighted throughout the manuscript, there is a clear advantage of this system over currently used high cost 3D or FACS-based models/methodologies. Simply the ability to have sustained PaCSC enrichment in a low cost 2D adherent system that is amenable to manipulation and does not require FACS is an enormous benefit.

2) When preparing the PDAC xenotransplant models for generating the 2D cultures, it is not explored how the different culture methods (Glu vs Gal) impact the initial tumor cell populations by selecting for different types of clones/cells present in the original tumor? Meaning PaCSCs are present in the original tumor and are selected by these novel methods or whether cells are responding to the in vitro culture system with a different source of energy and it is an adaptation that makes cells to express and behave in a certain way but that has nothing to do with what happen in vivo in the original tumor.

Authors comment: We first established the PDX-derived cell cultures from the different PDXs. After the cultures are established, passaged, and determined to be murine fibroblast free and “clean”, only then are cells placed in Gluc and Gal to test the effects of different carbon sources on CSC enrichment. Therefore, there was no selection process. Likewise, by first establishing the culture and then changing the carbon source, we are accurately studying how a mixed non-CSC/CSC population

reacts to changes in carbon sources via plasticity and adaptation to glycolysis deprivation.

Nevertheless, to understand what happens to PaCSCs and non-PaCSCs during the culture process in Gluc and Gal, we sorted the Autofluorescent-negative and CD133-negative population away from the bulk population of cells and placed the negative fractions in galactose and in glucose. Regardless of the carbon source, CSC-biomarker-negative cells gave rise to CSC-biomarker-positive cells (**Supplementary Figure 2a-b**). However, at times > 4 d post-sorting, a sharp decrease in the percentage of CSC-biomarker-positive cells was observed in Gluc-CC, while in Gal-CC the percentages either remained constant or increased, and at d 11 were ~2-fold higher than Gluc-CC (**Supplementary Figure 2a-b**). This fold-difference coincided with an ~2-fold decrease in cell viability in Gal-CC, confirming that PaCSC enrichment coincides with non-PaCSC cell death (**Supplementary Figure 2c**). Importantly, when CD133+ sorted cells were directly cultured in galactose, cell viability was not affected (**Supplementary Figure 2d**). Taken together, we concluded that in galactose-containing media 1) PaCSCs survive, 2) a large percentage of non-PaCSCs die, but 3) a smaller percentage of non-PaCSCs convert into PaCSCs (i.e. plasticity). The sum of these biological consequences results in a PaCSC-enriched culture. We have not been able to determine the mechanism(s) behind this plasticity, but discuss some possibilities in the re-written Discussion section.

Minor points

1) The data generated in Fig 1, related to the first section of results, comes from two of the PDAC-xenotransplant models that are described for the first time in the second section of results. This is a bit confusing. Can the authors describe the source of material (4 PDAC-xenotransplant models for identifying the CD133+ cells with high mitochondrial activity, already in the first section of results? Also, it will be more actuated to use the term PDAC-xenotransplant models, since this is the source for the cell cultures, rather than PDAC, that could drive to the mistake of assuming that cell cultures are generated from fresh PDAC tissue directly from patients.

Authors comment: All of the necessary information regarding the tumors are detailed in the Materials and Method's section, and references are provided. Secondly, while the terms used, such as PDX-derived are common, we have double checked the manuscript that ensure that there are no misleading terms or labels in the text, figures or corresponding legends.

2) The data generated in Figure 1C-G, how many hours cells are in culture from the sorting (for CD133 and Fluo) to the analysis?

Authors comment: Five days. This detail has been included.

3) The conclusion from Figure 1G that PaSCS (CD133+ cells) promote division because there are more nuclei counted (page 4 row 23) is difficult to agree with, considering the information provided in first paragraph of results and Figure legend. If cells were plated for counting nuclei, how many hours were in culture? Were the same number of sorted CD133+ ad CD133- neg cells plated? All this information is not indicated in Figure legend or result sections.

Authors comment: As noted above for Reviewer no.2. we agree with the Reviewer that this data was confusing and the use of DAPI as a marker of nuclear DNA dilution

may be questionable. Thus, since this observation is not of extreme relevance, we have removed it from the paper so as not to confuse the readers.

4) Figure 2A-2B, what PaSCS model the data belongs to (PANC185, PANCA6L?) It is not indicated in Figure legend or results section.

Authors comment: Shown in Panel A are representative plots, but not from one specific PDX-derived culture. This is indicated in the legend. In Panel B is the sum of all the data across the 4 PDX-derived cultures.

5) In Figure 2E it is indicated that PaSCS in Gal-CC had more self-renewal capacity than in Glu-CC, but when showing pictures of the spheroids (Figure 2F) the authors reported that the Spheres in Glu-CC were bigger. Can authors exclude that cells in Glu-CC condition have more capacity to fuse forming less spheres, but larger? Some cell counting after 7 or 14 days in culture it is needed to reach such conclusion. The data presented in Figures 2G and H does not seem enough to support the conclusion from Figure 2E, since this comparison is biased. The sphere experiment shows the capacity of cells to divide (or proliferate) and generate either the same cell type of different cell type. Therefore, self-renewal is not the right feature to measure in a sphere formation assay.

Authors comment: We apologize that the use of 7 and 14 days confused the review. Since 7 days is the time required to achieve a full-fledged sphere, we used 7 and 14 interchangeably with 1st and 2nd generation spheres. This has been corrected to avoid further confusion and we now state 1st and 2nd generation spheres.

We also disagree with the reviewer's comments about cells "fusing". Even if two cells fused, they would occupy the same space in the sphere, thus the size of the sphere would not change, only the absolute number of cells that make up the sphere. Rather, we believe that the smaller spheres are due to the enhanced slow-cycling "quiescent" state of the cells in Gal versus Gluc.

6) In Fig2I, expression levels for Nanog, Klf4 or Oct4 are not identified as markers for cancer stem cells form pancreatic cancers in pancreatic cancers (Zhao et al. Cancer Transl Med. 2017; 3(3): 87–95). Can authors evaluate the expression levels for pancreatic adult stem cell markers as PDX or ALDH1A1, together with SOX9 (already evaluated)?

Authors comment: Unfortunately, we completely disagree with the reviewer. The expression of the pluripotency-associated genes (*KLF4*, *NANOG*, *SOX2*, *OCT3/4*) is a hallmark of PaCSCs. A fact that we (Nat Methods. 2014 Nov;11(11):1161-9, Cancer Res. 2014 Dec 15;74(24):7309-20, Gut. 2015 Dec;64(12):1921-35, Cell Metab. 2015 Oct 6;22(4):590-605, Cancer Res. 2016 Aug 1;76(15):4546-58) and others have published countless numbers of times. In fact, the review cited by the reviewer states that the pluripotency-associated gene Oct3/4 is associated with CSCs.

7) In Suppl Fig4, it is presented the data for chemoresistance. In order to evaluate if drug screening assays performed with this novel in vitro model, a classical parameter to determine the dynamic range of the assay such Z' values are missing.

Authors comment: We are not evaluating a drug screening platform, but rather have performed a drug sensitivity experiment. A Z' value would only be necessary to calculate if we wanted to show the efficiency of the assay as a platform for low to high throughput screening of compound libraries. For merely showing the chemosensitivity of Gal-CC compared to Glu-CC, it is not necessary to calculate a Z' value.

Reviewers' Comments:

Reviewer #1:

Remarks to the Author:

In this revised study, Valle, et al. do well to address my two previous mechanistic concerns. 1) The data with direct comparison to "traditionally" sorted cancer stem cells will be useful to researchers in the field, so as to set experimental expectations. 2) With regard to the inquiry on the interchangeability of stem cells vs bulk, the observation that CD133⁻ cells can convert to CD133⁺ stem-like cells is curious. This, however, is not pursued, and noted as being outside of the scope of this study. I certainly appreciate the authors' point of view, and I agree that this study has done well to rigorously characterize the galactose-derived cells. These, I am convinced, are alike to traditionally-sorted cells. However, nearly all of this data is phenotypic characterization of already known properties of pancreatic cancer stem cells (see work from Heeschen, Draetta, and others). As such, while an exhaustive characterization, and comforting independent confirmation of pancreatic cancer stem cell properties, in its current form it lacks novel mechanistic insight into the how and why pancreatic cancer stem cells are enriched in mitochondrial features/functions.

Reviewer #2:

Remarks to the Author:

The authors have addressed most of my comments. I have only one major and a few minor critiques. Once these are addressed I look forward to seeing this work published. I understand that some of these may not be addressable due to the SARS-CoV-2 pandemic but hope that the data are available, or that these critiques (particularly the one major critique) can be addressed reasonably and rapidly once research institutions reopen.

Major

1. Bioluminescence imaging should be confirmed with histology. The authors have already confirmed the number of cells in distant organs by flow (Fig. 7f, g) but should confirm the luminescence data from the organs using histology (H&E or NuMA staining). If this is consistent with the bioluminescence data, this figure will be quite strong.

Minor

1. Fig. 1b. The black arrows indicating mitochondria are hard to see against the background of the CD133⁺ electron micrograph. Please outline them in white or otherwise increase their contrast with the background.
2. Fig. 2f. Did the authors quantitate the size of the spheres? It would be nice to have this to validate the claim of sphere size in line 170.
3. Supplementary Figure 2 a-d. Please mark the points that have significant differences between glucose- and galactose-grown Panc185 cells, or an n.s. if not significant. This was already done in multiple panels in Figures 5 and 6.
4. Why are the tumours in Fig. 2k larger at 1000 cells/injection than 5000 cells/injection? Please provide a scale bar for the tumour pictures.
5. The authors have demonstrated that PDAC cells increase both baseline and spare OCR, and have increased membrane potential, but do not demonstrate increased carbon flow into the TCA cycle using LC-MS or NMR-based metabolomics. If the authors have these data they should include it but if not they should at least mention that direct confirmation of increased carbon flux into the mitochondria could be validated in this manner.
6. There are some typos in the manuscript (e.g. Fig. 7e, organ analysis instead of organs analysis [sic]) but these will easily be addressed during editing.

Reviewer #3:

Remarks to the Author:

This reviewer was involved in the revision of this manuscript when it was submitted at Nature Method. Reading the answers of the Authors to my specific questions, I feel the main concerns were addressed. Reading also the answers addressing the points of the other two Reviewers, also has helped me to understand how the new version of the manuscript was generated.

The major claim of the manuscript is to described a new 2D in vitro system, by using galactose as main carbon source, for long-term enrichment of pancreatic CSCs. This 2D in vitro model generates a relevant model for studying not only pancreatic CSC, but any other CSC model. Addressing the comments of the previous revisions process, the claim of the manuscript is clear and the results support the conclusions. Therefore, this reviewer is positive about the work presented by the authors and have a positive opinion for its publication.

Response to Referees

NCOMMS-19-99997B

We thank all the Reviewers for their comments, and we are pleased that we have adequately addressed the vast majority of the comments raised in the initial review process.

Please find below our point-by-point response.

Reviewer #1 (Comments to the Author):

In this revised study, Valle, et al. do well to address my two previous mechanistic concerns. 1) The data with direct comparison to “traditionally” sorted cancer stem cells will be useful to researchers in the field, so as to set experimental expectations. 2) With regard to the inquiry on the interchangeability of stem cells vs bulk, the observation that CD133- cells can convert to CD133+ stem-like cells is curious. This, however, is not pursued, and noted as being outside of the scope of this study. I certainly appreciate the authors’ point of view, and I agree that this study has done well to rigorously characterized the galactose-derived cells. These, I am convinced, are alike to traditionally-sorted cells. However, nearly all of this data is phenotypic characterization of already known properties of pancreatic cancer stem cells (see work from Heesch, Draetta, and others). As such, while an exhaustive characterization, and comforting independent confirmation of pancreatic cancer stem cell properties, in its current form it lacks novel mechanistic insight into the how and why pancreatic cancer stem cells are enriched in mitochondrial features/functions.

Author’s response: We are happy that we have adequately addressed the Reviewer’s requests with the new experimental data added to the manuscript. We do respectfully disagree that “nearly all of this data is phenotypic characterization of already known properties of pancreatic cancer stem cells”. We have provided RNAseq and cytokine array analysis of galactose-cultured cells, and provide herein one of the first in-depth in vitro and in vivo analyses of CSC immune evasion properties associated with pancreatic cancer stem cells. Altogether, our system has allowed us to rigorously demonstrate that the metabolism of PaCSCs is intimately linked to their stem, slow-cycling, metastatic and immuno-evasive capacities. While more mechanistic studies are certainly warranted, we feel that a convincing and meticulous characterization of the system was first needed before more mechanistic future studies can be carried out. Rest assured that we are pursuing some very interesting avenues of research with the galactose-based PaCSCs culture system. In summary, we thank the reviewer for his/her time and input, and we feel that the manuscript is much improved as a direct result.

Reviewer #2 (Comments to the Author):

The authors have addressed most of my comments. I have only one major and a few minor critiques. Once these are addressed I look forward to seeing this work published. I understand that some of these may not be addressable due to the SARS-CoV-2 pandemic but hope that the data are available, or that these critiques (particularly the one major critique) can be addressed reasonably and rapidly once research institutions reopen.

Author’s response: We are happy that we have adequately addressed the Reviewer’s requests with the new experimental data added to the manuscript.

Major

1. Bioluminescence imaging should be confirmed with histology. The authors have already confirmed the number of cells in distant organs by flow (Fig. 7f, g) but should confirm the luminescence data from the organs using histology (H&E or NuMA staining). If this is consistent with the bioluminescence data, this figure will be quite strong.

Author’s response: For the experiment in question, we opted to use m-Cherry labeled human PDAC cells cultured with the respective carbon source (glucose vs galactose) as a rigorous tool to visualize and quantify the human PDAC cells in the respective organs at the indicated time points. For the small organs, including spleen, pancreas and lungs, all of the

Response to Referees

NCOMMS-19-99997B

organs were processed for cytometry in order to rigorously and accurately determine the percentage of mCherry cells in each organ, which for some organs hovered around 1%. Thus, unfortunately, we do not have blocks that could be stained to evaluate the presence of human cells at the histological level. We do have, however, have FFPE blocks with liver tissue and have performed, as requested by the reviewer, anti-human Cytokertin 19 staining to detect human cells in the mouse liver. As shown in the **new Supplementary Figure 7a-b**, the incidence of clusters of cytokertin19-positive cells is much higher in mice injected with Gal-CC versus Glu-CC at 3 months. Likewise, for several mice, large metastatic areas were observed and documented. Indeed, biofluorescence imaging and flow cytometry analysis are more sophisticated and rigorous techniques compared to histology, the latter of which can be quite qualitative and subjective. Nonetheless, we understand that visual confirmation of human cells in the mouse organs is very compelling, and we hope that the images provided convinces Reviewer no.2 that Gal-CC cells are more invasive and have a higher capacity to survive in circulation and colonize distant organs.

We feel that with these images, repeating the in vivo studies is not necessary. To generate all the samples the reviewer requests, we would have to launch a new intravenous injection experiment that would last 3 months. Our animal facility only just recently authorized the importation of commercially available mice. It is important to point out that the new organs resected and embedded for IHC would not correlate with the IVIS and cytometry analyses presented in the paper. Likewise, we should also consider that mice would be used solely to generate IHC images that are no more or better informative than the data already presented, thus such an experiment would go against the 3Rs. In summary, we feel that new in vivo experiments are not warranted in light of the images we have been able to provide, which correlate with the data already presented in the manuscript.

Minor

1. Fig. 1b. The black arrows indicating mitochondria are hard to see against the background of the CD133+ electron micrograph. Please outline them in white or otherwise increase their contrast with the background.

Author's response: We agree with the reviewer and have outlined the black arrows with white as suggested.

2. Fig. 2f. Did the authors quantitate the size of the spheres? It would be nice to have this to validate the claim of sphere size in line 170.

Author's response: We do not have access to a CASY cell counter that can efficiently quantitate the size of the spheres, but we have included a scale bar to highlight the appreciable differences in size between the Gluc-CC and Gal-CC spheres.

3. Supplementary Figure 2 a-d. Please mark the points that have significant differences between glucose- and galactose-grown Panc185 cells, or an n.s. if not significant. This was already done in multiple panels in Figures 5 and 6.

Author's response: We apologize for the oversight and significance is now shown for panels a-d of Supplementary Figure 2, as well as Panel i of Supplementary Figure 1.

4. Why are the tumours in Fig. 2k larger at 1000 cells/injection than 5000 cells/injection? Please provide a scale bar for the tumour pictures.

Author's response: This is a logical question. Tumors were allowed to develop for approximately 10-14 weeks; however, if a mouse in a dilution group developed an ulcerated tumor that warranted sacrifice of that mouse, all of the mice (Glu-CC and Gal-CC) in that dilution group were sacrificed in order to obtain the number of tumors for all mice at the exact same time. For the dilutions 50,000, 10,000 and 5,000, mice were sacrificed at 10 weeks as Gluc-CC tumors typically ulcerated or tumor burden impeded the health of mice. For the 1,000-cell dilution group, no tumors formed in the Gluc-CC group, nor did the Gal-CC tumors ulcerate; therefore, mice were taken out to 14 weeks. We have added these details

Response to Referees

NCOMMS-19-99997B

to the figure (noted as “end point”) and the Methods section (**Page 18, lines 578-581**) to avoid confusion. Lastly, as requested, we have provided a scale bar for the tumor pictures.

5. The authors have demonstrated that PDAC cells increase both baseline and spare OCR, and have increased membrane potential, but do not demonstrate increased carbon flow into the TCA cycle using LC-MS or NMR-based metabolomics. If the authors have these data they should include it but if not they should at least mention that direct confirmation of increased carbon flux into the mitochondria could be validated in this manner.

Author’s response: Unfortunately, we do not have these data, but as requested by the Reviewer, we have mentioned on **Page 8, lines 220-221** that direct confirmation of increased carbon flux into the mitochondria (i.e. TCA cycle) should be validated LC-MS or NMR-based metabolomics.

6. There are some typos in the manuscript (e.g. Fig. 7e, organ analysis instead of organs analysis [sic]) but these will easily be addressed during editing.

Author’s response: We thank the reviewer for catching these typos. We have gone through the entire manuscript and corrected these and additional typos.

Reviewer #3 (Comments to the Author):

This reviewer was involved in the revision of this manuscript when it was submitted at [REDACTED]. Reading the answers of the Authors to my specific questions, I feel the main concerns were addressed. Reading also the answers addressing the points of the other two Reviewers, also has helped me to understand how the new version of the manuscript was generated.

The major claim of the manuscript is to describe a new 2D in vitro system, by using galactose as main carbon source, for long-term enrichment of pancreatic CSCs. This 2D in vitro model generates a relevant model for studying not only pancreatic CSC, but any other CSC model.

Addressing the comments of the previous revisions process, the claim of the manuscript is clear and the results support the conclusions. Therefore, this reviewer is positive about the work presented by the authors and has a positive opinion for its publication.

Author’s response: We are pleased that we have adequately addressed the Reviewer’s requests and main concerns. We appreciate the Reviewer’s constructive criticisms and are happy that the reviewer “is positive about the work presented by the authors and has a positive opinion for its publication.”

Reviewers' Comments:

Reviewer #2:

Remarks to the Author:

The data in Supplementary Figure 7a and b show an increase in ck19-positive cells, although it has not been quantitated. The remainder of the minor points are addressed in the text or explained reasonably.

The authors have addressed my comments to the best of their ability and the manuscript should be ready for publication.

Response to Referee no2

NCOMMS-19-99997C

Below, please find a point-by-point response to the final comments from Reviewer no.2 and to the outstanding requests and format issues requested by the Editorial Office:

REVIEWER NO.2

The data in Supplementary Figure 7a and b show an increase in ck19-positive cells, although it has not been quantitated. The remainder of the minor points are addressed in the text or explained reasonably.

The authors have addressed my comments to the best of their ability and the manuscript should be ready for publication.

Authors' response: We thank the reviewer for the time dedicated to the review of this manuscript, which we feel is much improved due to the peer review process and now ready for publication.